# Molecular architecture of the human tRNA ligase complex

**Alena Kroupova[1], Fabian Ackle[1], Igor Asanović[2], Stefan Weitzer[2], Franziska M Boneberg[1], Marco Faini[3], Alexander Leitner[3], Alessia Chui[1], Ruedi Aebersold[3], Javier Martinez[2], Martin Jinek[1]\***

[1]Department of Biochemistry, University of Zurich, Zurich, Switzerland; [2]Max Perutz Labs, Vienna BioCenter (VBC), Vienna, Austria; [3]Department of Biology, Institute of Molecular Systems Biology, ETH Zurich, Zurich, Switzerland

**Abstract** RtcB enzymes are RNA ligases that play essential roles in tRNA splicing, unfolded protein response, and RNA repair. In metazoa, RtcB functions as part of a five-subunit tRNA ligase complex (tRNA-LC) along with Ddx1, Cgi-99, Fam98B, and Ashwin. The human tRNA-LC or its individual subunits have been implicated in additional cellular processes including microRNA maturation, viral replication, DNA double-strand break repair, and mRNA transport. Here, we present a biochemical analysis of the inter-subunit interactions within the human tRNA-LC along with crystal structures of the catalytic subunit RTCB and the N-terminal domain of CGI-99. We show that the core of the human tRNA-LC is assembled from RTCB and the C-terminal alpha-helical regions of DDX1, CGI-99, and FAM98B, all of which are required for complex integrity. The N-terminal domain of CGI-99 displays structural homology to calponin-homology domains, and CGI-99 and FAM98B associate via their N-terminal domains to form a stable subcomplex. The crystal structure of GMP-bound RTCB reveals divalent metal coordination geometry in the active site, providing insights into its catalytic mechanism. Collectively, these findings shed light on the molecular architecture and mechanism of the human tRNA ligase complex and provide a structural framework for understanding its functions in cellular RNA metabolism.

**\*For correspondence:**
jinek@bioc.uzh.ch

**Competing interest:** The authors declare that no competing interests exist.

## Editor's evaluation

Kroupova and colleagues present the manuscript 'Molecular architecture of human tRNA ligase complex,' which describes the first detailed dissection of the structure and assembly of the essential, multi-subunit tRNA ligase complex. The authors present, alongside crystal structures of the Rtcb catalytic subunit and the N-terminus of the associated CGI-99 subunit, a comprehensive deletion analysis and mass spectrometry investigation of the composition and assembly of the entire hetero-oligomeric assembly, identifying a novel subassembly between the CGI-99 and FAM98B subunits. The study is elegant, beautifully presented and written, easily followed, and interesting, providing a high-quality and important dissection of this essential complex.

## Introduction

RNA ligases play critical roles in various cellular processes including tRNA splicing (*Lopes et al., 2015*; *Phizicky and Hopper, 2010*; *Popow et al., 2012*; *Yoshihisa, 2014*), unfolded protein response (UPR) (*Bashir et al., 2021*; *Filipowicz, 2014*), RNA repair (*Burroughs and Aravind, 2016*), and generating U6/LINE-1 chimeric RNA retrotransposition substrates (*Moldovan et al., 2019*). Canonical ligases, including bacteriophage T4 RNA ligase 1 or *Saccharomyces cerevisiae* tRNA ligase Trl1, are ATP-dependent enzymes that catalyze the formation of a 3',5'-phosphodiester linkage between 3'-hydroxyl

(3′-OH) and 5′-phosphate (5′-P) RNA termini (*Banerjee et al., 2019*; *El Omari et al., 2006*; *Greer et al., 1983*; *Gu et al., 2016*; *Nandakumar et al., 2006*; *Peebles et al., 1979*; *Peschek and Walter, 2019*; *Unciuleac et al., 2015*). In tRNA splicing and other processes requiring the joining of RNA ends with a 2′,3′-cyclic phosphate (2′,3′>P) and a 5′-hydroxyl (5′-OH), these ligases operate as part of a multistep mechanism including separate phosphatase and kinase domains (*Banerjee et al., 2019*; *Chan et al., 2009*; *Remus et al., 2017*; *Sawaya et al., 2003*; *Wang et al., 2006*; *Xu et al., 1990*). By contrast, the RtcB ligases are GTP-dependent enzymes that catalyze the direct joining of RNA strands with terminal 2′,3′>P and 5′-OH (*Chakravarty et al., 2012*; *Tanaka and Shuman, 2011*). In contrast to the mechanism of canonical RNA ligases, in which the source of the splice-junction phosphate group is the ATP cofactor, RtcB ligases incorporate the substrate-derived 2′,3′>P into the resulting 3′,5′-phosphodiester bond (*Filipowicz and Shatkin, 1983*; *Popow et al., 2011*).

RtcB ligases are present throughout all domains of life with notable absences in plants and some fungi, including the model organisms *S. cerevisiae* and *Schizosaccharomyces pombe* (*Dantuluri et al., 2021*; *Popow et al., 2012*). Although their catalytic mechanism is conserved, their substrates and consequently their cellular functions vary considerably (*Englert et al., 2011*; *Popow et al., 2011*; *Popow et al., 2012*). In archaea and eukaryotes, RtcB enzymes catalyze the ligation of tRNA exon halves to produce mature tRNAs upon cleavage of precursor tRNA transcripts by the tRNA splicing endonuclease (*Calvin and Li, 2008*; *Li et al., 1998*; *Peebles et al., 1983*; *Trotta et al., 1997*). Furthermore, in eukaryotes, RtcB ligases also function as part of the UPR to catalyze splicing of *XBP1* mRNA after its stress-induced cleavage by IRE1 (inositol-requiring enzyme 1) (*Jurkin et al., 2014*; *Kosmaczewski et al., 2014*; *Lu et al., 2014*). In turn, bacterial RtcB ligases are nonessential enzymes involved in RNA repair, for example, during rescue of ribosomal RNAs cleaved by the MazF ribonuclease of the MazE-MazF toxin-antitoxin system activated upon stress (*Manwar et al., 2020*; *Temmel et al., 2017*).

The RNA ligation mechanism of RtcB enzymes consists of three nucleotidyl transfer steps. First, a nucleophilic attack of an invariant histidine residue in the active site (His428 in human RTCB, His404 in *Pyrococcus horikoshii* RtcB, PhRtcB, and His337 in *Escherichia coli* RtcB, EcRtcB) on the α-phosphate moiety of guanine-5′-triphosphate (GTP) results in a covalently linked RtcB-GMP intermediate with the concomitant release of inorganic pyrophosphate. Second, the terminal 2′,3′-cyclic phosphate of the 5′ exon is hydrolyzed to 3′-phosphate, which subsequently carries out a nucleophilic attack on the phosphate group of RtcB-GMP conjugate to form an activated RNA-(3′)pp(5′)G intermediate. Finally, the activated 3′ end of the 5′ exon undergoes nucleophilic attack by the 5′-OH of the 3′ exon, resulting in exon ligation and release of GMP (*Chakravarty and Shuman, 2012*; *Chakravarty et al., 2012*; *Desai et al., 2013*; *Englert et al., 2012*).

Crystallographic and biochemical studies of PhRtcB have identified interactions responsible for positioning the 3′ exon and the 5′-OH nucleophile in the final strand-joining step of the ligation reaction (*Banerjee et al., 2021*; *Maughan and Shuman, 2016*); in contrast, the exact positioning of the 5′ exon in the active site remains to be determined. The ligation reaction of RtcB enzymes is dependent on the presence of divalent metal ions, consistent with the observation of two active site-bound Mn²⁺ ions in the crystal structures of PhRtcB (*Das et al., 2013*; *Tanaka et al., 2011*; *Tanaka and Shuman, 2011*). Some RtcB orthologs, including human RTCB and PhRtcB, require Archease, a dedicated protein cofactor that facilitates the initial guanylation of RtcB and promotes its multiple turnover (*Desai et al., 2014*; *Popow et al., 2014*). This is thought to involve a conserved metal binding site in Archease that might influence the positioning of the divalent metal cations in the active site of RtcB (*Desai et al., 2015*). The catalytic activity of human RTCB was recently shown to be sensitive to copper-mediated oxidative inactivation under aerobic conditions (*Asanović et al., 2021*). The inactivation is counteracted by the oxidoreductase PYROXD1, which protects the active site of RTCB by forming a complex with RTCB in the presence of NAD(P)H (*Asanović et al., 2021*).

The human RtcB ligase ortholog RTCB, also referred to as HSPC117 or FAAP, is part of an essential heteropentameric tRNA ligase complex (tRNA-LC), along with the DEAD-box helicase DDX1, as well as FAM98B, CGI-99 (also known as RTRAF or hCLE), and ASHWIN (ASW) (*Popow et al., 2011*). RTCB is the only subunit of tRNA-LC that is essential and sufficient for RNA ligation, but it requires the protein cofactor Archease for multiple-turnover catalysis (*Popow et al., 2014*). DDX1 enhances the RTCB-catalyzed RNA ligation via its RNA-dependent ATPase activity (*Popow et al., 2014*), but it is also involved in other cellular processes including DNA double-strand break repair (*Li et al., 2016*; *Li et al., 2008*), microRNA maturation (*Han et al., 2014*), regulation of insulin translation (*Li et al.,*

*2018*), or regulation of viral replication including HIV-1 (*Edgcomb et al., 2012*; *Fang et al., 2004*), foot-and-mouth disease virus (*Xue et al., 2019*), infectious bronchitis virus (*Xu et al., 2010*), and SARS-CoV-2 (*Kamel et al., 2021*). Notably, it remains unclear whether these cellular functions are mediated by DDX1 in isolation or within the tRNA-LC. While CGI-99, FAM98B, and ASW have no effect on the RNA ligation activity of tRNA-LC (*Popow et al., 2011*), CGI-99 was shown to function as a transcription modulator in viral replication (*Huarte et al., 2001*; *Rodriguez-Frandsen et al., 2016*; *Rodriguez et al., 2011*). However, its mechanism of action, as well as the functions of FAM98B and ASW, is unknown.

Despite previous studies on archaeal and bacterial RtcB enzymes (*Chakravarty et al., 2012*; *Das et al., 2013*; *Desai et al., 2013*; *Englert et al., 2011*; *Englert et al., 2012*; *Maughan and Shuman, 2016*; *Okada et al., 2006*), our understanding of the mechanism of human RTCB function in the context of the tRNA-LC is incomplete. Here, we present a biochemical analysis of the inter-subunit interactions of the human tRNA ligase complex, along with crystal structures of the N-terminal domain of CGI-99 and the catalytic subunit RTCB in complex with guanosine-5′-monophosphate (GMP). These results provide key insights into the molecular architecture of the human tRNA ligase complex and its catalytic mechanism.

## Results

### RTCB, DDX1, FAM98B, and CGI-99 are essential for the formation of the core tRNA-LC

The human tRNA-LC was initially discovered to consist of five subunits: RTCB, DDX1, FAM98B, CGI-99, and ASW (*Figure 1A*; *Popow et al., 2011*). To identify subunits required for the integrity of tRNA-LC, we performed coexpression experiments using insect cells infected with recombinant baculovirus encoding the full-length tRNA-LC subunits RTCB, DDX1, FAM98B, CGI-99, and ASW. A stable, catalytically active complex could be obtained by coexpression of all five subunits and affinity purification (*Figure 1—figure supplement 1A and B*).

Next, we performed deletion analysis to define subunits essential for the formation of the complex using an approach that combined affinity purification from baculovirus-infected insect cells coupled with fluorescence detection. To be able to unambiguously identify the presence of each subunit, DDX1 and FAM98B were fused to mCherry and green fluorescent protein (GFP), respectively. Additionally, RTCB was tagged with a hexahistidine (His$_6$) tag, while CGI-99 carried a streptavidin-binding (StrepII) tag and ASW was fused N-terminally with the maltose-binding protein (MBP). Affinity purifications from insect cells infected with expression constructs lacking one of the subunits revealed that deletion of any of RTCB, DDX1, FAM98B, or CGI-99 resulted in failure of the remaining four subunits to form a stable complex (*Figure 1B*). Only ASW, a 26 kDa, intrinsically disordered protein with no conserved domains other than a leucine-rich region and two nuclear localization signals (*Patil et al., 2006*), could be omitted without compromising complex assembly. These results indicate that the integrity of tRNA-LC is dependent on the simultaneous presence of RTCB, DDX1, FAM98B, and CGI-99, suggesting that these subunits form a stable core of tRNA-LC. This conclusion is consistent with the evolutionary conservation of tRNA-LC subunits, whereas DDX1, FAM98B, and CGI-99 display similar phylogenetic distribution, ASW is absent from many organisms where the other subunits of tRNA-LC are present (*Popow et al., 2012*). Furthermore, affinity purifications of MBP-tagged ASW or StrepII-tagged CGI-99 revealed that in the absence of either DDX1 or RTCB, ASW forms a complex together with CGI-99 and FAM98B, but not with either of these subunits alone (*Figure 1B*). Additionally, CGI-99 and FAM98B were able to form a stable subcomplex in the absence of all other tRNA-LC subunits (*Figure 1B*). Taken together, these results suggest that the interaction of ASW with tRNA-LC is dependent on the CGI-99:FAM98B subcomplex and is likely mediated through the CGI-99:FAM98B interface.

Previous studies showed the conserved cofactor Archease to be essential for the guanylation of RTCB and its enzymatic turnover (*Popow et al., 2014*). Although RTCB binds Archease with nanomolar affinity, Archease does not co-purify with RTCB when the latter is affinity-purified from HEK293 cells (*Popow et al., 2014*), suggesting that Archease only transiently interacts with the tRNA-LC. To test whether Archease interacts stably with the tRNA-LC in vitro, we analyzed the interaction between recombinant purified Archease and tRNA-LC (containing subunits RTCB, DDX1, FAM98B, and CGI-99)

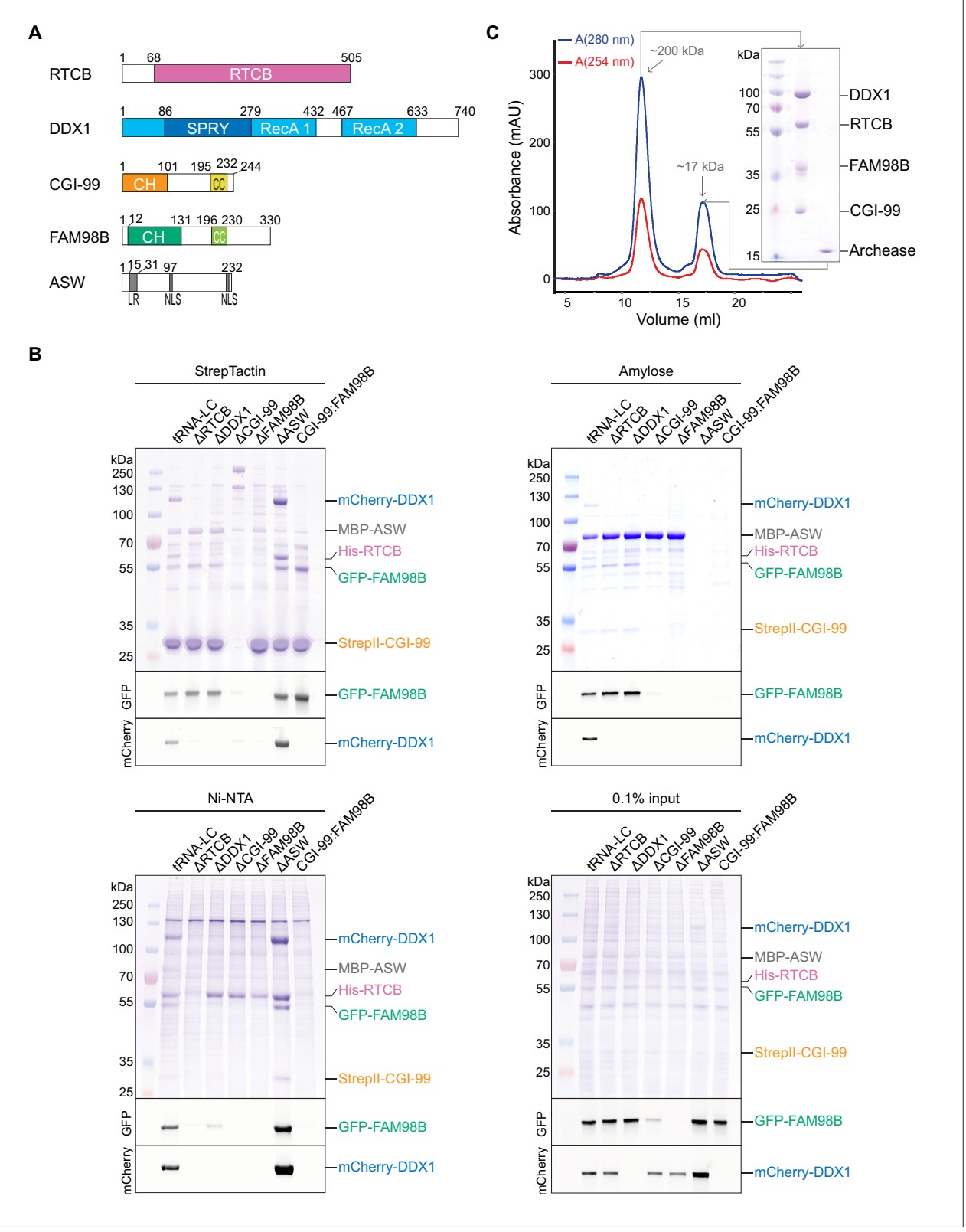

**Figure 1.** The core tRNA ligase complex consists of RTCB, DDX1, FAM98B, and CGI-99. (**A**) Domain composition of tRNA-LC subunits. RecA, RecA-like domain; CH, calponin homology-like domain; CC, coiled coil; LR, leucine-rich region; NLS, nuclear localization signal. (**B**) Deletion analysis of tRNA-LC subunits. StrepTactin (upper left), Amylose (upper right), and Ni-NTA (bottom left) affinity pull-down assays using lysates of Sf9 cells expressing full-length tRNA-LC subunits (affinity tags as indicated in the figure). The input (bottom right) and bound fractions were analyzed by SDS-

*Figure 1 continued*

PAGE and visualized by in-gel GFP (middle gel) or mCherry (bottom gel) fluorescence followed by Coomassie Blue staining (upper gel). tRNA-LC = RTCB:DDX1:FAM98B:CGI-99:ASW, omitted subunits in the deletion constructs are indicated (Δ). (**C**) Size-exclusion chromatography interaction analysis of the core tRNA-LC (RTCB:DDX1:FAM98B:CGI-99) and Archease. SDS-PAGE analysis of the elution peak components is shown in the inset. The estimated molecular weights of the peaks based on their elution volumes are indicated.

The online version of this article includes the following figure supplement(s) for figure 1:

**Figure supplement 1.** The recombinantly expressed tRNA-LC is catalytically active.

by size-exclusion chromatography (SEC; *Figure 1C*). In agreement with previous studies, tRNA-LC and Archease eluted in separate peaks, indicating that Archease does not form a stable complex with the tRNA-LC in vitro.

## Molecular topology of tRNA-LC

To investigate the molecular architecture and subunit connectivity of tRNA-LC, affinity-purified samples of both the five-subunit full complex and the four-subunit core complex were analyzed using cross-linking coupled to mass spectrometry (MS) to identify subunit regions involved in inter-subunit interactions. To this end, purified complexes were treated with disuccinimidyl suberate (DSS) to form covalent cross-links between lysine residues within a distance of approximately 35 Å, and cross-linked peptides were identified by MS. Analyzing two experimental replicates for each complex, we identified 103 or 143 validated intra-subunit and 104 or 141 validated inter-subunit cross-links for the five-subunit full complex and 156 or 146 validated intra-subunit and 102 or 92 validated inter-subunit cross-links for the four-subunit core complex (with a false discovery rate of <5%). Overall, this experiment revealed an extensive network of intra- (*Figure 2—figure supplement 1A*) and inter-subunit cross-links (*Figure 2*, *Figure 2—figure supplement 1B*). 66 of the inter-subunit cross-links involving RTCB, DDX1, CGI-99, and FAM98B were detected in both the four-subunit and five-subunit complex samples. The C-terminal region of DDX1 formed numerous cross-links with the remaining subunits of the five-subunit complex, with the exception of FAM98B, suggesting that the interaction of DDX1 with the remaining subunits within tRNA-LC is mediated by the C-terminal region (*Figure 2*). For RTCB and FAM98B, the cross-links also mapped mostly to their respective C-terminal regions, while CGI-99 cross-links were spread throughout its primary structure. ASW formed cross-links with all remaining subunits, particularly via its N-terminal region. A notable reduction of cross-links between the C-terminal regions of DDX1 and FAM98B was observed in the five-subunit full complex (*Figure 2A*) as compared to the four-subunit core complex (*Figure 2—figure supplement 1B*), suggesting that ASW is positioned between DDX1 and FAM98B or modulates their conformations. It is important to note, however, that formation of cross-links between protein regions depends not only on their spatial proximity but also on their conformational flexibility and the availability of lysine residues within them (*Tüting et al., 2020*).

## CGI-99 and FAM98B form a heterodimeric subcomplex via their N-terminal regions

Based on the observation that CGI-99 and FAM98B form a heterodimeric subcomplex (*Figure 1B*) and the detection of extensive cross-links between these tRNA-LC subunits (*Figure 2A*), we set out to probe this interaction in detail. Structural modeling using Phyre2 (*Kelley et al., 2015*) and PCOILS (*Lupas et al., 1991*) servers predicted that both CGI-99 and FAM98B comprised N-terminal calponin homology (CH)-like domains followed by putative C-terminal coiled-coil regions (*Figure 1A*; *Schou et al., 2014*). To verify this experimentally, we determined the crystal structure of the N-terminal domain of CGI-99 comprising residues Phe2-Asp101$^{CGI-99}$ at a resolution of 2.0 Å (*Table 1*). The protein construct adopts a globular fold connected via a flexible linker to an N-terminal α-helix comprising residues Phe2–Asn18$^{CGI-99}$, which is stabilized in different conformations in the two molecules present in the asymmetric unit by crystal packing interactions (*Figure 3A*). The closest structural homologs identified by the DALI server (*Holm, 2020*) include the CH domains of the human kinetochore protein NDC80 (PDB ID: 2ve7, root mean square deviation [rmsd] of 3.6 Å over 74 Cα atoms) and the *Chlamydomonas reinhardtii* intraflagellar transport protein IFT54 (PDB ID: 5fmt, rmsd 3.1 Å over 71 Cα atoms) (*Figure 3B*, *Figure 3—figure supplement 1A*), as predicted previously (*Schou et al., 2014*).

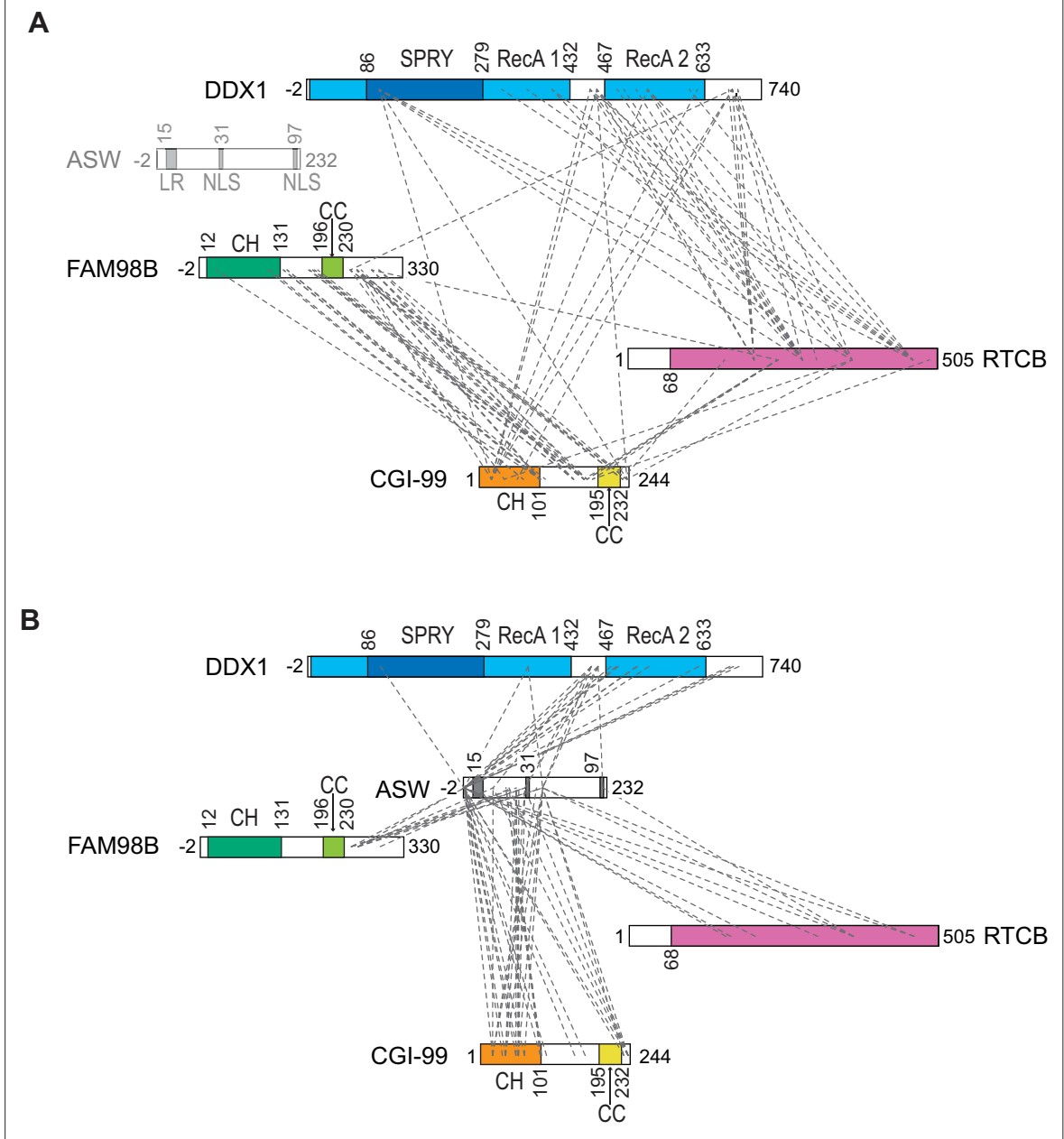

**Figure 2.** Cross-linking mass spectrometry analysis of the inter-subunit interactions within the five-subunit tRNA-LC (RTCB:DDX1:FAM98B:CGI-99:ASW). (**A**) Inter-subunit cross-links between RTCB, DDX1, CGI-99, and FAM98B subunits within the five-subunit full tRNA-LC are shown as dashed lines. (**B**) Inter-subunit cross-links between ASW and the remaining subunits of the full tRNA-LC. Domain schematic as in *Figure 1A*. The dashed lines indicate cross-links observed at least once in two replicates.

The online version of this article includes the following figure supplement(s) for figure 2:

**Source data 1.** Cross-links identified in the cross-linking and mass spectrometric (XL-MS) analysis of the full tRNA-LC.

**Source data 2.** Cross-links identified in the cross-linking and mass spectrometric (XL-MS) analysis of the core tRNA-LC.

**Figure supplement 1.** Cross-linking mass spectrometry analysis of tRNA-LC.

However, the N-terminal domain of CGI-99 also exhibits close structural homology with non-CH domains such as the helical N-terminal domain of *E. coli* Lon protease (PDB ID: 3ljc, rmsd 2.7 Å over 80 Cα atoms) (*Figure 3—figure supplement 1B*). Together, these observations confirm that the N-terminal domain of CGI-99 adopts a calponin homology-like fold.

**Table 1.** Crystallographic data collection and refinement statistics.

| | RTCB-GMP product complex (PDB ID: 7P3B) | CGI-99 N-terminal domain (PDB ID: 7P3A) | |
| --- | --- | --- | --- |
| | | Native | S-SAD |
| *Data collection* | | | |
| Space group | $P4_12_12$ | $P6_1$ | $P6_1$ |
| *Cell dimensions* | | | |
| a, b, c (Å) | 96.91, 96.91, 227.14 | 91.71, 91.71, 52.94 | 91.83, 91.83, 53.01 |
| $\alpha$, $\beta$, $\gamma$ (°) | 90, 90, 90 | 90, 90, 120 | 90, 90, 120 |
| Wavelength (Å) | 1.0000 | 1.0084 | 2.0173 |
| Resolution (Å) | 48.99–2.30 (2.38–2.30) | 45.86–2.00 (2.07–2.00) | 44.11–2.24 (2.32–2.24) |
| Total reflections | 1287302 (123,580) | 349,085 (33,359) | 808,522 (11,139) |
| Unique reflections | 48,937 (4769) | 17,246 (1725) | 23,759 (1159) |
| $R_{merge}$ (%) | 34.2 (325.7) | 14.2 (153.2) | 53.9 (71.8) |
| $R_{pim}$ (%) | 6.8 (64.8) | 3.2 (35.4) | 8.5 (31.0) |
| $I/\sigma I$ | 11.5 (1.1) | 18.6 (1.90) | 25.6 (2.5) |
| $Cc(1/2)$ | 0.997 (0.548) | 0.999 (0.74) | 0.999 (0.684) |
| Completeness (%) | 99.9 (99.4) | 99.9 (99.9) | 99.5 (95.7) |
| Redundancy | 26.3 (25.9) | 20.2 (19.3) | 34.0 (9.6) |
| *Refinement* | | | |
| Resolution (Å) | 48.99–2.30 | 45.86–2.00 | |
| No. of reflections | 48,912 | 17,235 | |
| $R_{work}/R_{free}$ | 0.192/0.220 | 0.197/0.224 | |
| *No. of non-hydrogen atoms* | | | |
| Protein | 7492 | 1688 | |
| Ligand/ion | 78 | 42 | |
| Water | 297 | 128 | |
| *B-factors* (Å$^2$) | | | |
| Protein | 45.2 | 39.4 | |
| Ligand/ion | 42.9 | 50.7 | |
| Water | 44.3 | 42.8 | |
| *R.m.s. deviations* | | | |
| Bond lengths (Å) | 0.023 | 0.012 | |
| Bond angles (°) | 1.68 | 1.11 | |
| *Ramachandran plot* | | | |
| % favored | 97.64 | 98.96 | |
| % allowed | 2.36 | 1.04 | |
| % outliers | 0 | 0 | |

Each dataset was collected from a single crystal. Values in parentheses are for highest-resolution shell.

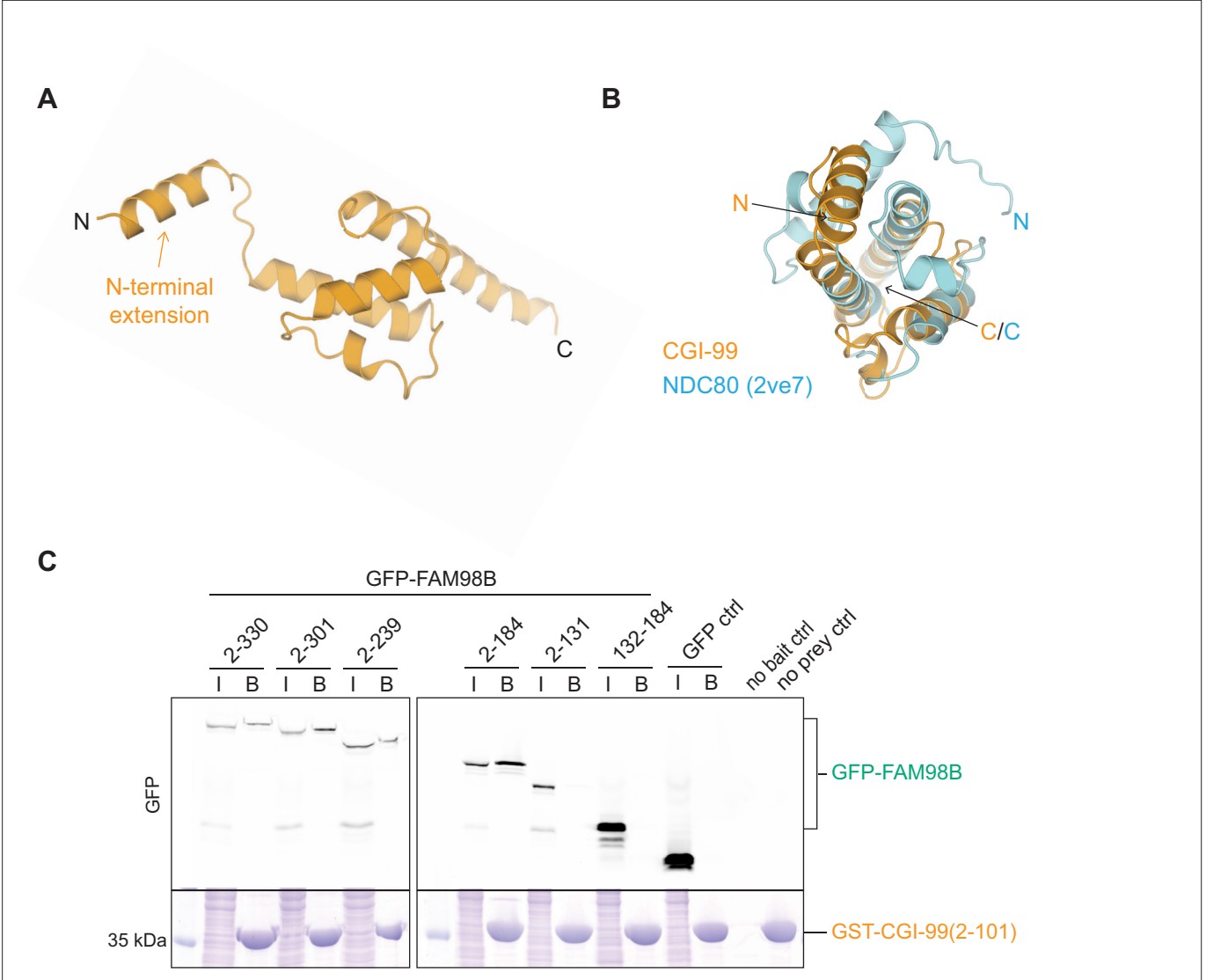

**Figure 3.** The interaction of CGI-99 and FAM98B is mediated via their N-terminal regions. (**A**) Crystal structure of the N-terminal region of human CGI-99 (residues 2–101). The N- and C-termini, as well as the N-terminal extension of CGI-99 (residues 2–18), are indicated. (**B**) DALI pairwise alignment of CGI-99(2-101) (yellow) with human NDC80 (blue, PDB ID: 2ve7). (**C**) Pull-down assays using pre-immobilized GST-CGI-99(2-101) with lysates from HEK293T cells transiently overexpressing HA-(StrepII)$_2$-GFP-FAM98B constructs. The input (I) and bound (B) fractions were analyzed by SDS-PAGE and visualized by in-gel GFP fluorescence (upper panel) followed by Coomassie Blue staining (lower panel). The 'no bait' control contained HEK293T lysate incubated with GSH resin in the absence of GST-CGI-99, 'no prey ctrl' contained GSH resin incubated with GST-CGI-99(2-101).

The online version of this article includes the following figure supplement(s) for figure 3:

**Figure supplement 1.** Analysis of the N-terminal domain of CGI-99 and the CGI-99:FAM98B interaction.

Although both CH domains and coiled-coil regions are known to mediate protein-protein interactions, the existence of a CGI-99:FAM98B complex and its functional relevance in vivo have not been reported to date. Interestingly, the overall level of overexpressed FAM98B was reduced in the absence of CGI-99, indicating that CGI-99 has a stabilizing effect on FAM98B in the cell. To pinpoint the interacting regions of CGI-99 and FAM98B, we performed co-precipitation assays. As FAM98B could not be expressed and purified in isolation, we instead expressed full-length and truncated FAM98B constructs as N-terminal GFP fusions in HEK293T cells and subsequently incubated cell lysates with purified GST-tagged CGI-99 constructs immobilized on a Glutathione Sepharose resin. The N-terminal domain of CGI-99 (Phe2-Asp101[CGI-99]) was able to co-precipitate a minimal FAM98B construct spanning residues Arg2-Lys184[FAM98B] (*Figure 3C, Figure 3—figure supplement 1C and*

*D*), indicating that the CGI-99:FAM98B interaction is dependent on both the N-terminal CH domain and a predicted beta-hairpin motif (residues Glu145-Lys184$^{FAM98B}$) in FAM98B, and the N-terminal CH domain in CGI-99. In contrast, the putative C-terminal coiled-coil regions of CGI-99 and FAM98B were dispensable for the interaction (*Figure 3—figure supplement 1D*). Taken together, these results indicate that the formation of the CGI-99:FAM98B heterodimer is mediated by the respective N-terminal regions of the subunits and suggest that the N-terminal CH-like domains of CGI-99 and FAM98B may comprise an independently functioning module within tRNA-LC.

## C-terminal regions of DDX1, FAM98B, and CGI-99 mediate the assembly of the core tRNA ligase complex

DDX1 is an 82 kDa DEAD-box helicase comprising two canonical RecA-like domains (RecA1 and RecA2), an atypical SPRY domain inserted in the N-terminal RecA1 domain and a C-terminal extension (*Figure 1A*; *Kellner and Meinhart, 2015*). The molecular functions of the SPRY domain and the C-terminal extension remain unknown. Based on the cross-linking/mass spectrometry analysis (XL-MS; *Figure 2*, *Figure 2—figure supplement 1B*), we reasoned that DDX1 interacts with the remaining subunits of tRNA-LC via its C-terminal region. To test this, we coexpressed in baculovirus-infected insect cells full-length RTCB, CGI-99, and FAM98B with a truncated DDX1 construct comprising residues Ser436-Phe740$^{DDX1}$, lacking the RecA1 and SPRY domains. Following affinity purification, SEC analysis confirmed integrity of the complex, indicating that the C-terminal region of DDX1, comprising the RecA2 domain and the C-terminal extension, is sufficient to support tRNA-LC assembly (*Figure 4—figure supplement 1A*). The purified complex was subsequently subjected to limited proteolysis with trypsin in order to identify exposed, protease-sensitive regions such as flexibly linked domains or unstructured loops. MS analysis of fragments separated by SEC revealed DDX1 cleavage, resulting in the loss of a fragment spanning residues Ser436-Arg694$^{DDX1}$ corresponding to the RecA2 domain (*Figure 4—figure supplement 1A and B*, *Figure 4—source data 1*). In light of the previously demonstrated requirement of DDX1 for tRNA-LC integrity, this result implies that the C-terminal extension of DDX1, corresponding to residues Ala695-Phe740$^{DDX1}$, is sufficient to mediate tRNA-LC assembly.

Guided by these results, we set out to identify the minimal subunit regions sufficient for tRNA-LC assembly. As with the initial deletion analysis of the full-length tRNA-LC subunits, we designed poly-promoter constructs for coexpression in baculovirus-infected Sf9 cells, with the following tagging scheme: His$_6$-RTCB, MBP-FLAG-DDX1, mCherry-CGI-99, and GFP-FAM98B. Each set of constructs contained N- or C-terminal truncations of one of the subunits with the others remaining full-length (*Figure 4A–C*). Co-precipitation experiments revealed that a minimal fragment of DDX1 corresponding to the C-terminal extension (residues Ala696-Phe740$^{DDX1}$) was sufficient for the assembly of the four-subunit core tRNA-LC. In contrast, neither a shorter C-terminal DDX1 construct (residues Leu729-Phe740$^{DDX1}$) nor a DDX1 construct containing the RecA1 and RecA2 domains but lacking the C-terminal extension (residues Ala2-Ala695$^{DDX1}$) were able to support tRNA-LC assembly. For both CGI-99 and FAM98B, we observed that their N-terminal domains, although sufficient for the CGI-99:FAM98B interaction (*Figure 3C*), were dispensable for tRNA-LC assembly (*Figure 4B and C*). Instead, a CGI-99 fragment spanning residues Ala195-Asp232$^{CGI-99}$ was essential and sufficient for the formation of the core complex. The minimal region of FAM98B that was essential and sufficient for tRNA-LC assembly could be mapped to a region spanning residues Asn200-Ser239$^{FAM98B}$. Notably, these regions are predicted to have a propensity for forming alpha-helical coiled coils (*Lupas et al., 1991*), while the C-terminal extension of DDX1 is predicted to form an alpha-helix.

To validate the subunit features sufficient for tRNA-LC assembly, we designed a 'minimal' complex comprising DDX1 residues Ala696-Phe740$^{DDX1}$, CGI-99 residues Leu102-Arg244$^{CGI-99}$, and FAM98B residues Asn200-Ser239$^{FAM98B}$, along with full-length RTCB. Coexpression of the subunit constructs in baculovirus-infected insect cells resulted in the assembly of a stable, catalytically active complex, as determined by affinity purification, SEC analysis, and in vitro RNA ligation assay (*Figure 4D*, *Figure 1—figure supplement 1B*). Of note, a fragment of CGI-99 comprising residues Leu102-Arg244$^{CGI-99}$ was necessary for the assembly of the minimal complex, as opposed to a shorter fragment (Ala195-Asp232$^{CGI-99}$) sufficient for complex assembly when DDX1 and FAM98B were full-length (*Figure 4B*). Taken together, these results indicate that the structural core of tRNA-LC is composed of RTCB together with the C-terminal alpha-helical regions of DDX1, CGI-99, and FAM98B, and suggest that DDX1, CGI-99, and FAM98B interact synergistically to form a structural platform that interacts

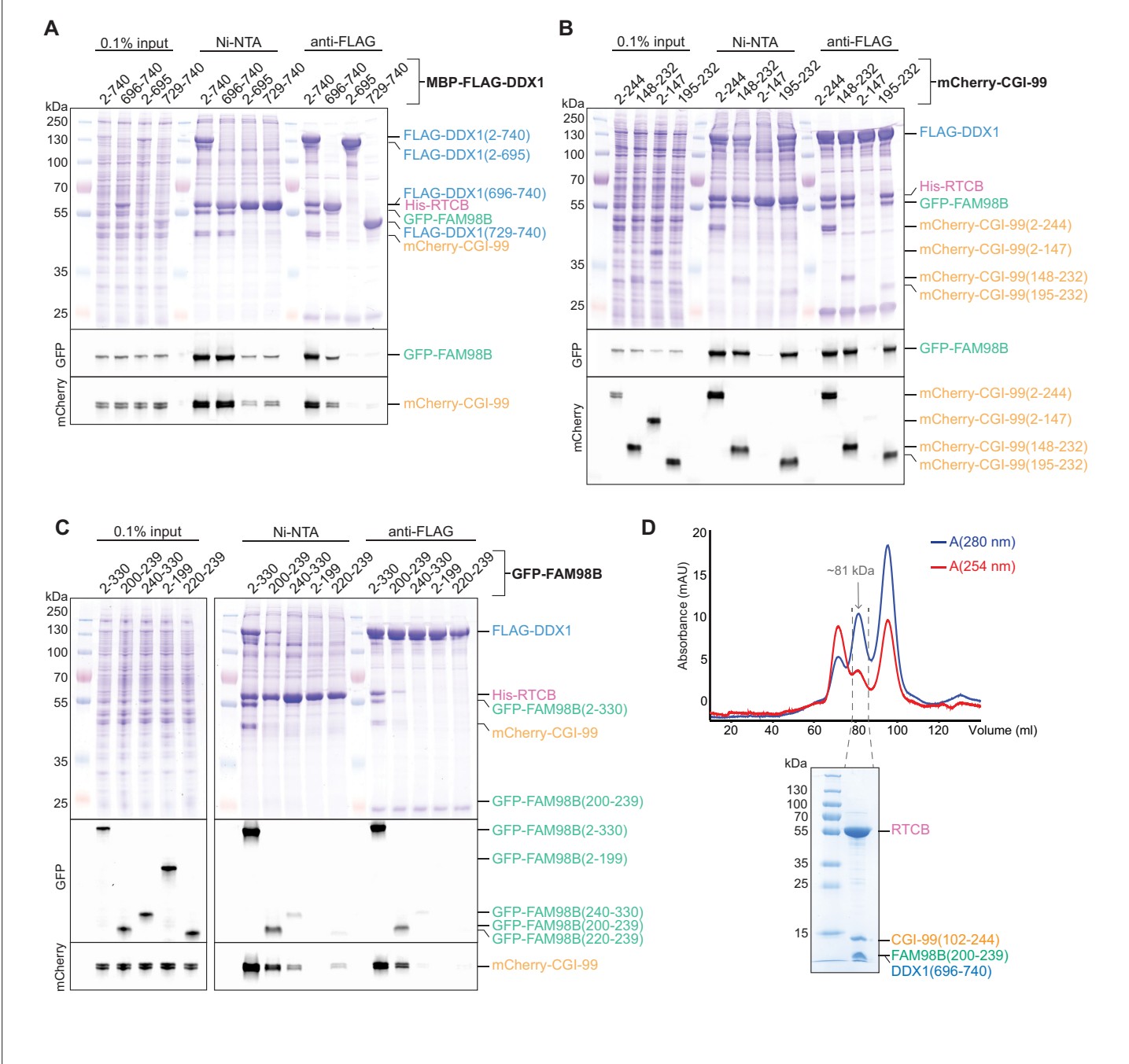

**Figure 4.** The C-terminal regions of DDX1, CGI-99, and FAM98B facilitate the formation of the core tRNA ligase complex. (**A–C**) Truncation analysis of the core tRNA-LC (RTCB:DDX1:FAM98B:CGI-99). Pull-down assays on lysates of Sf9 cells expressing constructs of the core tRNA-LC with truncated DDX1 (**A**), CGI-99 (**B**), or FAM98B (**C**). The input and bound fractions were analyzed by SDS-PAGE and visualized by in-gel GFP (middle panel) or mCherry (bottom panel) fluorescence followed by Coomassie Blue staining (upper panel). (**D**) Top: size-exclusion chromatography purification of the minimal tRNA-LC, RTCB:DDX1(696-740):FAM98B(200-239):CGI-99(102-244). The estimated molecular weight of the peak according to its elution volume is indicated. Bottom: SDS-PAGE analysis of the final sample. The collected fractions are indicated by gray dashed lines.

The online version of this article includes the following figure supplement(s) for figure 4:

**Source data 1.** Mass spectrometry analysis of protein fragments resulting from limited proteolysis of RTCB:DDX1(436-740):FAM98B:CGI-99.

**Figure supplement 1.** Mass spectrometry analysis of RTCB:DDX1(436-740):FAM98B:CGI-99 subjected to limited proteolysis.

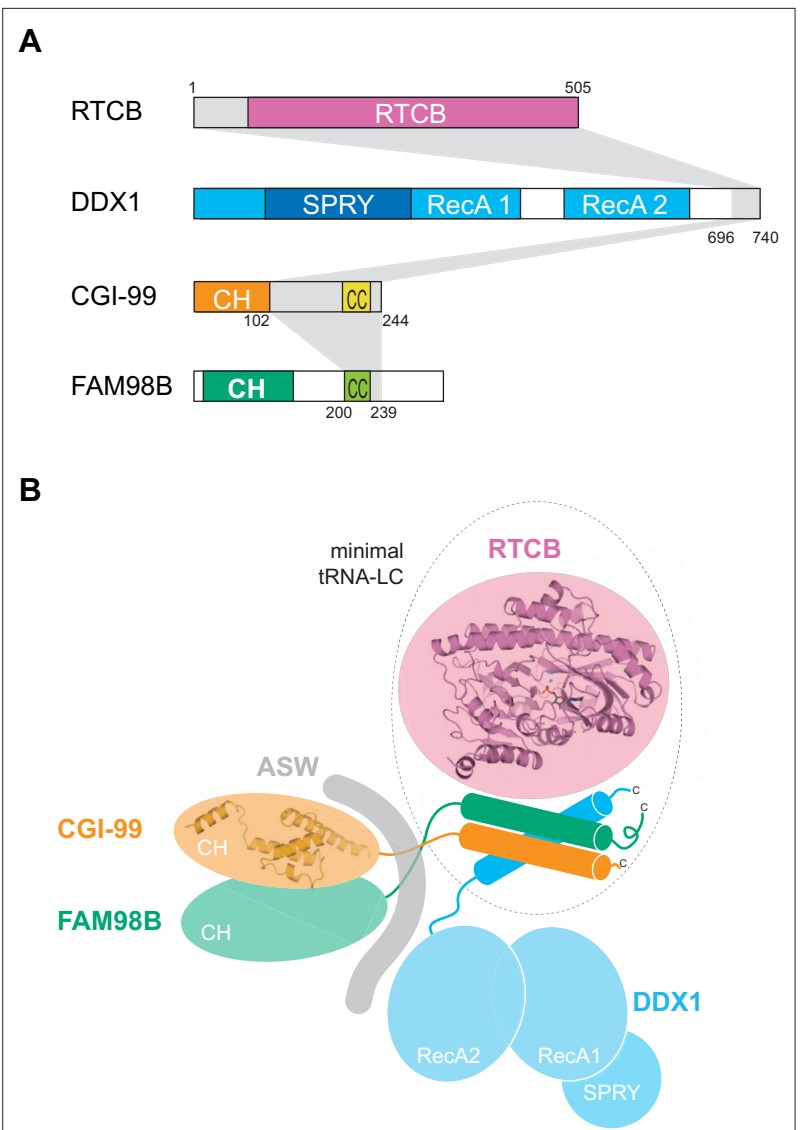

**Figure 5.** The architecture of the human tRNA ligase complex. (**A**) Subunit regions required for the formation of the minimal tRNA-LC. Domain composition is shown as in *Figure 1A*, minimal regions are highlighted in gray. (**B**) Schematic representation of the molecular architecture of tRNA-LC.

with the catalytic domain of the RTCB ligase subunit (*Figure 5*). Interestingly, the ligase activity of the minimal complex was substantially increased as compared with the four- and five-subunit complexes containing full-length proteins, and comparable to the activity of RTCB in isolation (*Figure 1—figure supplement 1B*). This suggests that the noncatalytic tRNA-LC subunits might negatively regulate the catalytic activity of RTCB via specific regions that are not involved in complex assembly. The activity of the complex was not increased in the presence of ATP, suggesting that the RNA-dependent ATPase activity of DDX1 is not involved in ligase inhibition (*Figure 1—figure supplement 1C*).

## Crystal structure of human RTCB reveals conserved fold and active site with distinct metal coordination geometry

While metazoan RtcB enzymes exist as part of the tRNA ligase complex, prokaryotic RtcB orthologs function as stand-alone enzymes (*Popow et al., 2012*). Unlike EcRtcB, which exhibits clear selectivity for $Mn^{2+}$ and is inhibited by $Co^{2+}$ (*Das et al., 2013*; *Desai and Raines, 2012*; *Tanaka et al., 2011*), human RTCB is highly active in the presence of either $Co^{2+}$ or $Mn^{2+}$ ions, and exhibits detectable activity in the presence of $Mg^{2+}$ or $Zn^{2+}$ (*Figure 6—figure supplement 1*). Crystallographic studies

of PhRtcB have provided insights into the catalytic mechanism of stand-alone RtcB enzymes, shedding light on the mechanisms of GTP-dependent activation, 3′ exon recognition, and divalent cation coordination (**Banerjee et al., 2021**; **Desai et al., 2013**; **Englert et al., 2012**; **Okada et al., 2006**). However, we currently lack structural information on the tRNA-LC-resident RtcB enzyme. To address this, we determined, at a resolution of 2.3 Å, the crystal structure of human RTCB in complex with a GMP molecule and two divalent cobalt ions bound in the active site (**Figure 6A**, **Table 1**).

The conserved fold of RTCB superimposes with PhRtcB (PDB ID: 4dwq) with rmsd of 1.4 Å over 474 Cα atoms (**Figure 6B**). Two adjacent loops found in the vicinity of the active site in RTCB are structurally distinct from PhRtcB: an N-terminal region spanning residues Leu45-Pro64$^{RTCB}$, which forms an extended α-helix and a loop as compared to a shorter loop (Lys37-Arg42$^{PhRtcB}$) in PhRtcB, and a segment comprising residues Arg436-Asp444$^{RTCB}$, which is only partially ordered (**Figure 6B**). To test whether the N-terminal RTCB extension is involved in the inter-subunit interactions within tRNA-LC, we coexpressed a truncated RTCB containing residues Pro64-Gly505$^{RTCB}$ with full-length DDX1, FAM98B, and CGI-99. Affinity purification revealed that the N-terminal extension of human RTCB is not essential for the formation of tRNA-LC (**Figure 6—figure supplement 2**).

The structure of GMP-bound RTCB likely mimics an on-pathway, product-bound state (**Figure 6A**). The GMP guanosine base makes hydrogen bonding interactions with the sidechains of Glu230$^{RTCB}$, Ser409$^{RTCB}$, and Lys504$^{RTCB}$. The ribose 2′ and 3′ OH groups are hydrogen-bonded with backbone nitrogens of Ala430$^{RTCB}$ and Gly431$^{RTCB}$, while the phosphate oxygen atoms interact with the sidechains of Asn226$^{RTCB}$ and His428$^{RTCB}$ (**Figure 6C**). The phosphate moiety is further coordinated by the Co$^{2+}$(A) ion. The GMP binding pocket is conserved with respect to the structure of PhRtcB in the guanylated state (PDB ID: 4dwq) with the exception of Glu470$^{RTCB}$, which, unlike in the PhRtcB structure, adopts a conformation incompatible with hydrogen bonding with N7 of the guanosine base (**Figure 6D**; **Englert et al., 2012**). The two Co$^{2+}$ ions, referred to as Co$^{2+}$(A) and Co$^{2+}$(B), adopt tetrahedral and octahedral coordination geometries, respectively (**Figure 6E**, left panel). Co$^{2+}$(A) is coordinated by His259$^{RTCB}$, His353$^{RTCB}$, Cys122$^{RTCB}$, and the phosphate moiety of GMP, whereas Co$^{2+}$(B) is coordinated by Asp119$^{RTCB}$, Cys122$^{RTCB}$, His227$^{RTCB}$, and three water molecules. Intriguingly, the metal coordination geometries in the RTCB-GMP structure differ from those adopted by Mn$^{2+}$ ions in the PhRtcB structures (**Figure 6E**, **Table 2**), in which the A-position ion exhibits octahedral (PDB ID: 4dwr) or trigonal bipyramidal (PDB ID: 4dwq) coordination arrangements, while the B-position ion is tetrahedrally coordinated (PDB ID: 4dwr). These observations indicate considerable structural plasticity of metal ion coordination and suggest that the coordination geometries might dynamically vary throughout the catalytic cycle of the enzyme to facilitate each step of the overall reaction mechanism.

## Discussion

The tRNA ligase complex is an essential factor for both tRNA biogenesis and *XBP1* mRNA splicing during the unfolded protein response in human cells (**Jurkin et al., 2014**; **Kosmaczewski et al., 2014**; **Lu et al., 2014**; **Popow et al., 2011**). Despite considerable progress, there remain gaps in our understanding of its cellular function and its molecular architecture, particularly with respect to the noncatalytic subunits of the complex. In this study, we provide insights into the inter-subunit interactions within tRNA-LC, as well as high-resolution structures of its catalytic subunit RTCB and the N-terminal CH domain of CGI-99.

Using coexpression experiments complemented with XL-MS, we show that a four-subunit core complex is assembled by the C-terminal regions of DDX1, CGI-99, and FAM98B together with the catalytic domain of the RTCB subunit. Each of the three noncatalytic subunits is required for the integrity of the complex, suggesting that they function synergistically to generate an interaction platform for RTCB. As the C-terminal regions of DDX1, FAM98B, and CGI-99 are predicted to be alpha-helical and, in the case of CGI-99 and FAM98B to form coiled-coils, we hypothesize that they associate to form a helical bundle (**Figure 5**). The C-terminal extension of DDX1 required for tRNA-LC assembly is located outside the helicase core comprising the tandem RecA1 and RecA2 domains and the SPRY domain. This suggests that the helicase (DDX1) and ligase (RTCB) modules of tRNA-LC are flexibly linked, which may serve to facilitate interactions with a diverse array of RNA substrates. The molecular architecture of tRNA-LC might also play a role in regulating the enzymatic activity of the ligase subunit RTCB, as suggested by our biochemical analysis of full-length and truncated complexes, and isolated RTCB.

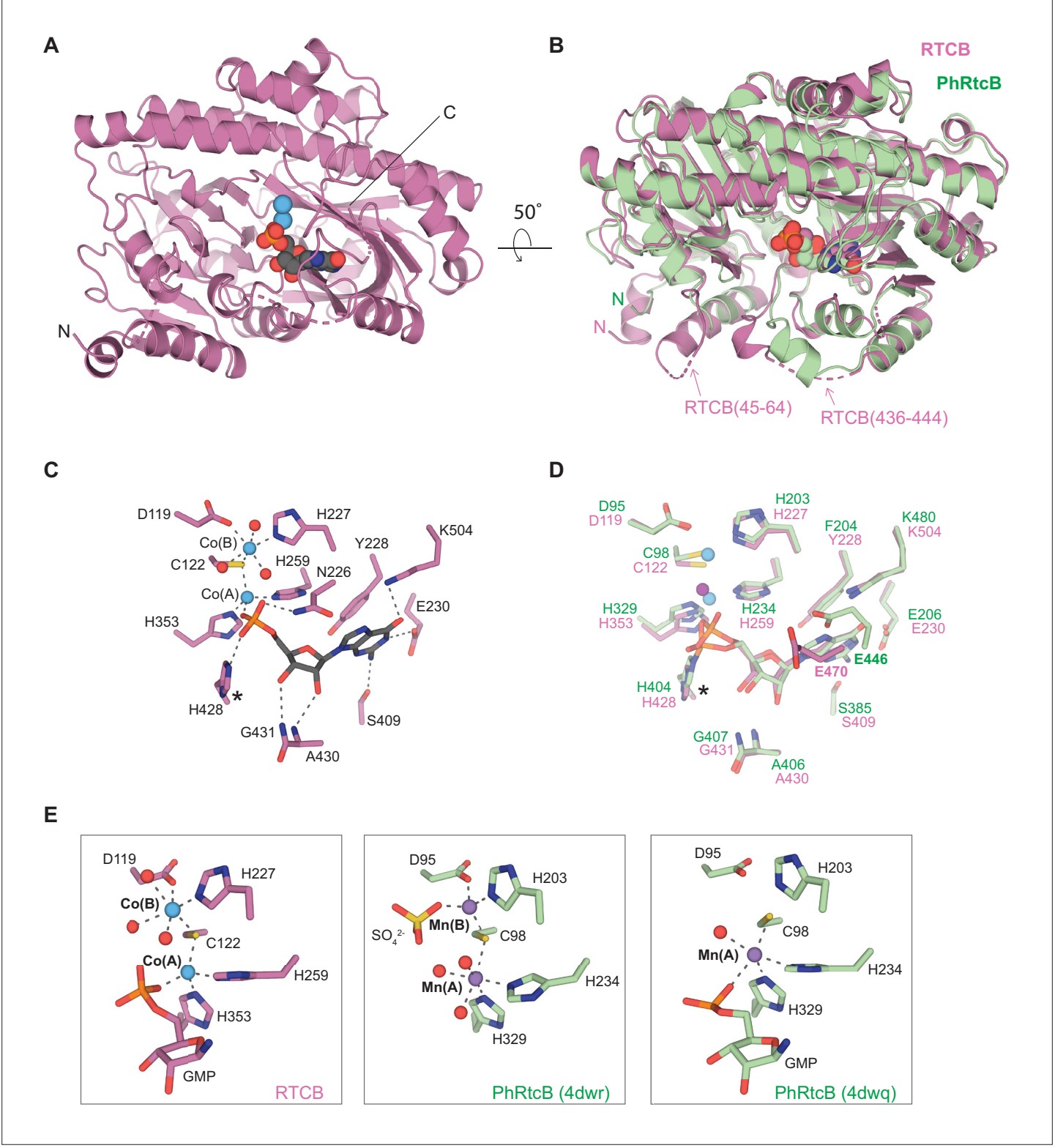

**Figure 6.** Crystal structure of human RTCB in complex with GMP and Co$^{2+}$. (**A**) Overall fold of RTCB. GMP is shown in space-fill representation, Co$^{2+}$ ions are shown as blue spheres. The N- and C-termini are indicated. (**B**) DALI superposition of the GMP-bound RTCB (pink) and PhRtcB (green, PDB ID: 4dwq) structures. The nonconserved regions spanning residues Leu45-Pro64$^{RTCB}$ and Arg436-Asp444$^{RTCB}$ are indicated. (**C**) Detailed view of the active site RTCB residues (shown as pink sticks). GMP is shown as gray sticks, Co$^{2+}$ ions as blue spheres and water molecules as red spheres. The hydrogen bonds and metal coordination bonds are shown as gray dashed lines. The histidine residue undergoing guanylation is labeled with an asterisk. (**D**) Detailed

*Figure 6 continued on next page*

*Figure 6 continued*

view of the active site of the superposed RTCB (pink) and PhRtcB (green, PDB ID: 4dwq). The $Co^{2+}$ and $Mn^{2+}$ ions are shown as blue and purple spheres, respectively. The histidine residues undergoing guanylation are labeled with an asterisk. (**E**) Detailed view of the metal coordination in the structures of GMP-bound RTCB (left panel), $SO_4^{2-}$-bound PhRtcB (PDB ID: 4dwr, middle panel), and guanylated PhRtcB (PDB ID: 4dwq, right panel). Color scheme as in panel (**D**).

The online version of this article includes the following figure supplement(s) for figure 6:

**Figure supplement 1.** Divalent metal selectivity of RTCB and tRNA-LC.

**Figure supplement 2.** N-terminal loop of RTCB is not essential for the formation of the core tRNA-LC.

In addition, the CGI-99 and FAM98B subunits of tRNA-LC form a stable heterodimer through an independent interaction interface involving their N-terminal domains. This heterodimer further serves as an interaction platform for the recruitment of ASW, the fifth tRNA-LC subunit. The existence of CGI-99:FAM98B or CGI-99:FAM98B:ASW subcomplexes in vivo has not been reported to date, and it is currently unclear whether they have a biological role independent from tRNA-LC. CGI-99 was previously shown to form a denaturation-resistant homodimer under certain conditions, although only monomeric CGI-99 is associated with the other subunits of tRNA-LC (*Pérez-González et al., 2014*). Consistently, we have not observed CGI-99 dimerization in vitro.

Our crystal structure of the N-terminal domain of CGI-99 confirms that it shares similarities with CH domains, which are typically found in a range of signaling and cytoskeletal proteins (*Gimona et al., 2002*; *Korenbaum and Rivero, 2002*; *Yin et al., 2020*). The CH1- and CH2-type CH domains occur in tandem in proteins such as α-actinin (*Shams et al., 2016*), β-III-spectrin (*Avery et al., 2017*), or dystrophin (*Singh and Mallela, 2012*) and mediate their binding to F-actin. Contrary to the tandem CH1-CH2 domains, CH3-type occur as stand-alone domains at the N-termini of proteins such as calponin. Apart from actin binding, CH domains have also been found to mediate interactions with microtubules (MT). The end-binding protein 1 (EB-1) regulates microtubule dynamics, with its N-terminal CH domain being essential for its MT interaction (*Hayashi and Ikura, 2003*). Similarly, the CH domains of NDC80 and NUF2 in the NDC80 kinetochore complex mediate MT binding, although unlike in EB-1 this is conditional on the formation of the NDC80-NUF2 heterodimer (*Ciferri et al., 2008*; *Wei et al., 2007*). Our structural analysis reveals that the CH-like domain of CGI-99 exhibits closest structural homology to the microtubule binding NDC80-like family of CH domains, as proposed by an earlier study (*Schou et al., 2014*). There is, however, no evidence to date to indicate that CGI-99 and FAM98B, or indeed tRNA-LC, interact with MTs or other cytoskeletal proteins. Nevertheless, tRNA-LC has been implicated in several cytoskeleton-related processes, including RNA transport along microtubules (*Kanai et al., 2004*) and regulation of cell-adhesion dynamics (*Hu et al., 2008*). CGI-99 was also found to interact with the tubulin binding protein PTPIP51 (protein tyrosine phosphatase interacting protein 51) (*Brobeil et al., 2012*) and the centrosomal protein ninein (*Howng et al., 2004*). Further studies will be necessary to understand whether the CH-like domain of CGI-99, perhaps together with

**Table 2.** Metal coordination geometries found in RtcB structures.

| Species | PDB ID | Chain | Metal | Ligand | Metal coordination geometry | |
| --- | --- | --- | --- | --- | --- | --- |
| | | | | | **Position A** | **Position B** |
| *Homo sapiens* | 7P3B | A, B | $Co^{2+}$ | GMP | Tetrahedral | Octahedral |
| | 4DWQ | A, B | $Mn^{2+}$ | GMP* | Trigonal bipyramidal | – |
| | | A | $Mn^{2+}$ | GMP* | Tetrahedral | Trigonal pyramidal |
| | 4IT0 | B | $Mn^{2+}$ | – | Tetrahedral | Tetrahedral |
| | | A | $Mn^{2+}$ | GTP$\alpha$S | Tetrahedral | Tetrahedral |
| | 4ISZ | B | $Mn^{2+}$ | GTP$\alpha$S | Trigonal bipyramidal | Tetrahedral |
| | 4DWR | A, B, C | $Mn^{2+}$ | $SO_4^{2-}$ | Octahedral | Tetrahedral |
| *Pyrococcus horikoshii* | 4ISJ | A, B | $Mn^{2+}$ | – | Tetrahedral | – |

*Denotes a GMP ligand covalently bound to the His404 of PhRtcB.

the putative CH N-terminal domain of FAM98B, facilitates a cytoskeleton-related cellular function of tRNA-LC or plays another functional role.

The crystal structure of human RTCB determined in this study reveals a product-bound state containing GMP and two divalent cobalt ions in the active site. The structure uncovers the disposition of the active site residues, contradicting previous predictions based on homology modeling (*Nandy et al., 2017*). The observed metal coordination geometries are also distinct from those found in the crystal structures of the apo and guanylated intermediate states of PhRtcB. The RTCB structure shows that the metal ion A is tetrahedrally coordinated, instead of adopting octahedral coordination as observed in the $Mn^{2+}$-bound PhRtcB structures. Unlike the prokaryotic RtcB orthologs, human RTCB is active in the presence of $Co^{2+}$ ions, suggesting that the $Co^{2+}$-bound RTCB structure represents an on-pathway state and not a catalytically inhibited complex. One possible explanation for the discrepancy between the RTCB and PhRtcB structures is that the divalent metal ion coordination geometries dynamically vary throughout the RtcB catalytic mechanism, thereby facilitating the individual reaction steps of enzyme guanylation, 5'-exon activation and exon ligation. Moreover, it is conceivable that active site positions A and B are occupied by nonidentical metal ions under physiological conditions in vivo. The question of metal ion selectivity of RtcB enzymes thus remains unresolved. Additional structural studies of intermediate catalytic states and different metal ion-bound complexes will thus be needed to uncover whether the observed plasticity of metal ion coordination in RtcB active site plays a role in the catalytic mechanism. Finally, both RTCB and the entire tRNA-LC were recently shown to be redox-regulated and to undergo oxidative inactivation in the presence of copper, suggesting that the binding of copper ions in the active site promotes oxidation of the active site cysteine (Cys122[RTCB]), thus precluding proper divalent metal coordination (*Asanović et al., 2021*). The NAD(P)H-dependent interaction of RTCB with PYROXD1 counteracts this process (*Asanović et al., 2021*). However, the precise chemical mechanism by which PYROXD1 protects RTCB is currently unknown. Furthermore, the RNA substrate binding mechanism of RTCB is likewise not fully understood. Previous mutational studies of *E. coli* and *P. horikoshii* RtcB enzymes have implicated conflicting sets of amino acid residues in exon RNA recognition (*Englert et al., 2012*; *Maughan and Shuman, 2016*). Although a recent crystal structure of PhRtcB bound to a 5'-OH 3'DNA oligonucleotide (*Banerjee et al., 2021*) revealed the residues involved in 3' exon binding, additional structures of RtcB enzymes bound to the 5' exon or to both exons simultaneously will be required to pinpoint the precise position of the 2',3'>P end, to identify the active site residues involved in catalyzing the hydrolysis of the 2',3'>P into a 3'-P intermediate, thus providing further insights into the latter steps of the ligation reaction beyond the initial guanylation of RtcB.

In conclusion, this work provides fundamental insights into the molecular architecture of the human tRNA-LC, paving the way for its further structural and mechanistic studies. These will shed light on specific molecular determinants of inter-subunit interactions and the ways in which they support the catalytic activity as well as putative noncatalytic functions of the complex, thereby advancing our understanding of tRNA-LC in metazoan RNA metabolism.

## Materials and methods
### Expression and purification of five-subunit tRNA-LC (RTCB:DDX1:FAM98B:CGI-99:ASW)

DNA encoding human RTCB (UniProt Q9Y3I0) and CGI-99 (UniProt Q9Y224) was each inserted into the UC Berkeley MacroLab 438A vector (gift from Scott Gradia, Addgene plasmid #55218) using ligation-independent cloning. DDX1 (UniProt Q92499), FAM98B (UniProt Q52LJ0), and ASW (UniProt Q9BVC5-1) were subcloned into the UC Berkeley MacroLab plasmids 438B (Addgene plasmid #55219), 438Rgfp (Addgene plasmid #55221), and 438C (Addgene plasmid #55220), respectively. The plasmids containing individual subunits were combined into a single baculovirus transfer plasmid using the MacroBac protocol (*Gradia et al., 2017*). The resulting plasmid encoded untagged RTCB; DDX1 with an N-terminal $His_6$ tag followed by a Tobacco Etch Virus (TEV protease cleavage site); ASW with an N-terminal $His_6$ tag followed by an MBP tag and a TEV protease cleavage site, FAM98B with an N-terminal StrepII-tag followed by a GFP tag followed by a TEV protease cleavage site; and untagged CGI-99. The recombinant baculovirus was generated using the Bac-to-Bac baculovirus expression system (Invitrogen), and the proteins were expressed in Sf9 insect cells (Thermo Fisher

Scientific, cat. no. 11496015; cell line was authenticated and tested for mycoplasma contamination by manufacturer, no further validation was done by the authors) infected at a density of $1.0 \times 10^6$ ml$^{-1}$ with P3 baculovirus stocks. The cells were harvested 60 h post-infection, resuspended in lysis buffer (20 mM HEPES pH 8.0, 150 mM NaCl, 5 mM imidazole, 0.1% Tween 20, supplemented with Roche cOmplete Protease Inhibitor Cocktail) and lysed by sonication. The lysate was clarified by centrifugation for 30 min at 30,000 × $g$ at 4°C, and the supernatant was applied to two 5 ml Ni-NTA Superflow cartridges (QIAGEN). The columns were washed with 20 mM HEPES pH 8.0, 500 mM NaCl, and 10–20 mM imidazole, and bound protein was eluted with buffer containing 20 mM HEPES pH 8.0, 500 mM NaCl, and 250 mM imidazole. The protein solution was loaded onto a 6 ml of StrepTactin Superflow resin (IBA) pre-equilibrated in Strep wash buffer (20 mM HEPES pH 8.0, 250 mM NaCl, 0.5 mM TCEP), the matrix was washed with Strep wash buffer and the protein was eluted with Strep wash buffer supplemented with 5 mM desthiobiotin. The fusion tag was removed by a His$_6$-tagged TEV protease during an overnight dialysis at 4°C against buffer containing 20 mM HEPES pH 8.0 and 250 mM NaCl. The dialyzed protein mixture was concentrated using a centrifugal filter (Amicon Ultra, MWCO 30 kDa, Sigma), and the complex was further purified by SEC (Superdex 200 increase 10/300, Cytiva, formerly GE Healthcare), eluting with 20 mM HEPES pH 7.5, 150 mM KCl, 0.5 mM TCEP. Peak fractions containing the five tRNA-LC subunits were pooled, concentrated to 2.7 mg ml$^{-1}$ using a centrifugal filter (Amicon Ultra, MWCO 30 kDa, Sigma), flash frozen in liquid nitrogen, and stored at –80°C.

## Expression and purification of four-subunit tRNA-LC (RTCB:DDX1:FAM98B:CGI-99)

DNA sequences encoding human RTCB (UniProt Q9Y3I0) and DDX1 (UniProt Q92499) were cloned into UC Berkeley MacroLab 5B vector (gift from Scott Gradia, Addgene plasmid #30122) with DDX1 inserted into site 1 to generate a fusion with an N-terminal His$_6$ tag followed by TEV protease cleavage site, and RTCB inserted into site 2 with no affinity tags. Similarly, DNA encoding CGI-99 (UniProt Q9Y224) was cloned into site 1 of the UC Berkeley MacroLab 5A vector (gift from Scott Gradia, Addgene plasmid #30121), while DNA sequence encoding FAM98B (UniProt Q52LJ0) N-terminally fused to StrepII tag, GFP, and TEV protease cleavage site (amplified from the UC Berkeley MacroLab 438-Rgfp vector, Addgene plasmid #55221) was cloned into site 2 of the same vector. Recombinant baculoviruses were generated using the Bac-to-Bac baculovirus expression system (Invitrogen), and the proteins were expressed in Sf9 insect cells infected at a density of $1.2 \times 10^6$ ml$^{-1}$ with P3 baculovirus stocks in a 1:1 (v/v) ratio. The cells were harvested 60 h post-infection, resuspended in lysis buffer (20 mM Tris pH 8.0, 150 mM NaCl, 5 mM imidazole, 0.4% Triton X-100, supplemented with Roche cOmplete Protease Inhibitor Cocktail) and lysed by sonication. The lysate was clarified by centrifugation for 30 min at 30,000 × $g$ at 4°C, and the supernatant was incubated with 6 ml HIS-Select Nickel Affinity Gel (Sigma), gently agitated at 4°C for 1 h. The solution was transferred to a glass column and the matrix was washed with 20 mM Tris pH 8.0, 500 mM NaCl, and 10 mM imidazole, and eluted with the same buffer containing 250 mM imidazole. The fusion tags were removed by incubation with His$_6$-tagged TEV protease during overnight dialysis at 4°C against a buffer containing 20 mM Tris pH 8.0 and 250 mM KCl. The protein solution was subsequently applied to a 5 ml Ni-NTA Superflow cartridge (QIAGEN), which was washed with dialysis buffer supplemented with increasing concentrations of imidazole (10–250 mM). The tRNA-LC-containing flow-through and wash fractions were collected, concentrated using a centrifugal filter (Amicon Ultra, MWCO 100 kDa, Sigma), and loaded on a SEC column (Superdex 200 16/600, Cytiva) equilibrated in 20 mM HEPES pH 8.0 and 250 mM KCl. Peak fractions containing the four tRNA-LC subunits were collected and concentrated to 6.6 mg ml$^{-1}$ using a centrifugal filter (Amicon Ultra, MWCO 100 kDa, Sigma). The sample was flash frozen in liquid nitrogen and stored at –80°C until further use.

## Expression and purification of Archease

DNA encoding a fragment of human Archease (UniProt Q8IWT0) spanning residues 27–167 was cloned into the UC Berkeley MacroLab 2M-T vector (gift from Scott Gradia, Addgene plasmid #29708) to express a fusion protein containing an N-terminal His$_6$ tag, MBP tag, and a TEV protease cleavage site. The fusion protein was expressed in BL21 (DE3) Rosetta2 *E. coli* cells induced with 0.2 mM IPTG at 18°C overnight. After harvesting, the cells were lysed in 20 mM Tris pH 8.0, 500 mM NaCl, and

5 mM imidazole, and the lysate was clarified by centrifugation for 30 min at 20,000 × $g$ at 4°C. The clarified lysate was applied to two tandemly connected 5 ml Ni-NTA Superflow cartridges (QIAGEN); the cartridges were washed with buffer containing 20 mM Tris pH 8.0, 500 mM NaCl, 10 mM imidazole, and bound protein was eluted with buffer containing 20 mM Tris pH 8.0, 500 mM NaCl, and 250 mM imidazole. The fusion tag was removed with His$_6$-tagged TEV protease during overnight dialysis at 4°C against 20 mM Tris pH 8.0, 250 mM KCl, and 5% glycerol. The sample was subsequently reapplied onto the Ni-NTA Superflow cartridges to remove the affinity tag and His$_6$-TEV protease. The flow-through fraction was concentrated using a centrifugal filter (Amicon Ultra, MWCO 10 kDa, Sigma) and purified by SEC (Superdex 75, Cytiva), eluting with buffer containing 20 mM HEPES pH 7.5, 250 mM KCl, and 1 mM DTT. Archease-containing peak fractions were concentrated to 43 mg ml$^{-1}$ using a centrifugal filter (Amicon Ultra, MWCO 10 kDa, Sigma), flash frozen in liquid nitrogen, and stored at –80°C.

## Interaction analysis of tRNA-LC and Archease

2.5 nmol of tRNA-LC was mixed with 5 nmol Archease and incubated on ice for 30 min. The mixture was loaded onto a Superdex 200 10/300 SEC column equilibrated in 20 mM HEPES pH 8.0, 250 mM KCl, and 1 mM DTT. The peak fractions were analyzed by sodium dodecyl sulfate-polyacrylamide gel electrophoresis (SDS-PAGE) stained with Coomassie Blue.

## Deletion analysis of tRNA-LC subunits

DNA encoding human RTCB (UniProt Q9Y3I0) was inserted into the UC Berkeley MacroLab 438B vector (Addgene plasmid #55219) using ligation-independent cloning, and DDX1 (UniProt Q92499), FAM98B (UniProt Q52LJ0), CGI-99 (UniProt Q9Y224), and ASW (UniProt Q9BVC5-1) were subcloned into the UC Berkeley MacroLab 438A (Addgene plasmid #55218) plasmid along with sequences encoding N-terminal affinity/fluorescence tags. The resulting plasmids encoded RTCB with an N-terminal His$_6$ tag followed by a TEV protease cleavage site; DDX1 with an N-terminal mCherry tag followed by a TEV protease cleavage site; FAM98B with an N-terminal GFP tag followed by a TEV protease cleavage site; CGI-99 with an N-terminal (StrepII)$_2$ tag followed by a PreScission protease cleavage site; and ASW with an N-terminal MBP tag followed by a PreScission protease cleavage site. The plasmids containing individual subunits were combined into a single baculovirus transfer plasmid using the MacroBac protocol (*Gradia et al., 2017*). The 'tRNA-LC' construct contained all five subunits, whereas the deletion constructs lacked each one subunit, as indicated. The CGI-99:FAM98B construct contained only the two subunits CGI-99 and FAM98B. Recombinant viruses were generated using the Bac-to-Bac baculovirus expression system (Invitrogen). The proteins were expressed by infecting 50 ml of Sf9 insect cells at a density of $1.0 \times 10^6$ ml$^{-1}$ with the above-described viruses. Infected cells were harvested by centrifugation at 500 × $g$ for 10 min at room temperature (RT). The cell pellet was frozen in liquid nitrogen and stored at –20°C until further use. Subsequently, the pellet was thawed and resuspended in 4.5 ml lysis buffer containing 20 mM HEPES pH 8.0, 150 mM NaCl, 0.1% Tween 20, 5 mM imidazole, supplemented with 1× Roche cOmplete Protease Inhibitor Cocktail. The cells were lysed by sonication and nucleic acids were digested by incubating with Benzonase Nuclease (125 units, Sigma) for 30 min at 4°C. The lysate was clarified by centrifugation at 8500 rpm, 4°C for 10 min and split into three parts. The first part was loaded onto 200 µl of Ni-NTA Superflow resin (QIAGEN) in a gravity column equilibrated with Ni wash buffer (20 mM HEPES pH 8.0, 150 mM NaCl, 15 mM imidazole, 0.1% Tween 20). The resin was washed with 4 × 1 ml Ni wash buffer and eluted with 250 µl (E1) followed by 50 µl (E2) of Ni wash buffer containing 250 mM imidazole. The second fraction of the clarified lysate was loaded onto 200 µl of Amylose Resin (New England Biolabs) equilibrated in Amy-Strep wash buffer (20 mM HEPES pH 8.0, 150 mM NaCl, 0.1% Tween 20) in a gravity column. The resin was washed with 3 × 1 ml Amy-Strep wash buffer and eluted with 200 µl (E1) and 50 µl (E2) of wash buffer supplemented with 10 mM maltose. The third fraction of the clarified lysate was incubated for 1 h at 4°C with 2 µl of MagStrep Type 3 XT Beads (IBA) equilibrated with Amy-Strep wash buffer. The supernatant was removed, and the resin was washed three times with 400 µl of Amy-Strep wash buffer. Bound proteins were eluted with 30 µl of 1× SDS loading dye (45 mM Tris pH 6.8, 10% glycerol, 5% SDS, 50 mM DTT, bromophenol blue). Samples for analysis by SDS-PAGE were taken from cleared lysate ( = input) and E2 elution fractions. Samples were resolved on 10% Mini-PROTEAN TGX Precast Protein Gels (Bio-Rad). The samples were not heated before loading to preserve the fluorescence of

the GFP and mCherry tags. Before staining, the gels were visualized using a Typhoon FLA 9500 fluorescence scanner (Cytiva) with the 473 nm and the 532 nm excitation laser settings to detect GFP and mCherry, respectively. The gels were subsequently stained by Coomassie Blue.

## RNA ligation assay

50 pmol RNA oligonucleotide derived from the firefly luciferase gene (5′-UCG AAG UAU UCC GCG UAC GU-3′, Dharmacon) were 3′ end-radiolabeled by incubation for 1 h at 16°C with 1.11 MBq [5′-$^{32}$P] cytidine-3′,5-bisphosphate (222 TBq mmol$^{-1}$, Hartmann Analytic) and 10 units T4 RNA ligase 1 (NEB) in a total reaction volume of 20 µl containing 15% (v/v) DMSO, 50 mM Tris-HCl pH 7.6, 10 mM MgCl$_2$, 10 mM β-mercaptoethanol, 200 µM ATP, 0.1 mg ml$^{-1}$ BSA. Labeling reactions were resolved by denaturing gel electrophoresis in a 15% polyacrylamide gel containing 8 M urea (SequaGel, National Diagnostics). 3′ end-labeled RNA was visualized by autoradiography and passively eluted from gel slices overnight at 4°C in 0.3 M NaCl. The RNA was subsequently precipitated by adding three volumes of ethanol and the RNA pellet resuspended in RNase-free water. The activity of recombinant ligase was measured using an RNA (circularization) ligation assay. Recombinant ligase and its essential co-factor Archease (*Popow et al., 2014*), at 50 nM or 200 nM equimolar concentrations, were incubated at 30°C with 5 nM 3′ end-labeled RNA substrate in a reaction buffer containing 115 mM KCl, 8 mM HEPES pH 7.4, 50 mM Tris-HCl pH 8.0, 8% glycerol, 0.1 mM GTP, 2 mM MnCl$_2$, 5.4 mM DTT, and (where indicated) 1.6 mM ATP. After given time points, the reaction was stopped by addition of 2xFA solution (1/1, v/v), composed of 90% formamide, 50 mM EDTA, 1 ng ml$^{-1}$ bromophenol blue, 1 ng ml$^{-1}$ xylene cyanol, and heated to 95°C for 3 min. The reaction products were separated on a denaturing 15% urea-polyacrylamide gel (SequaGel) and detected by autoradiography.

## Divalent metal selectivity assay

Recombinant core tRNA-LC or RTCB (25 nM) and Archease (200 nM) were preincubated with EDTA (10 µM) on ice for 5 min, in the buffer containing 30 mM HEPES/KOH pH 7.4, 100 mM KCl, 5 mM MgCl$_2$, 40 mM AEBSF, 10% glycerol, and 1% NP40. Next, metal ions (2 mM) were added, and the sample was incubated on ice for 1 h. Separately, a reaction cocktail was generated by mixing (1:2, v/v) radiolabeled oligonucleotide substrate with a solution containing 5 mM TCEP, 100 mM KCl, 7.5 mM ATP, 0.5 mM GTP, RNasin Ribonuclease Inhibitor (0.2%, v/v) and 65% (v/v) glycerol. The reaction cocktail was mixed with the RTCB/metal sample at a 3:2 (v/v) ratio and incubated for 30 min at 30°C in a total volume of 5 µl. The reaction was then stopped by adding equal volume of 2xFA solution composed of 90% formamide, 50 mM EDTA, 1 ng ml$^{-1}$ bromophenol blue, 1 ng ml$^{-1}$ xylene cyanol, and heated to 95°C for 1 min. The reaction products were separated on a denaturing 15% urea-polyacrylamide gel (SequaGel) and detected by autoradiography. ImageQuant software was used to quantify the band intensities, and substrate conversion rate was calculated as the fraction of the substrate and the sum of substrate and product band intensities.

## Pull-down assay of the CGI-99:FAM98B subcomplex

DNA sequences encoding N-terminally GFP-fused and truncated FAM98B constructs were cloned into the pcDNA5/FRT/TO/nHASt vector (Invitrogen). The resulting FAM98B truncations contained an N-terminal HA-(StrepII)$_2$-GFP tag. 2 × 10$^6$ cells HEK293T cells were transfected with 2 µg plasmid DNA using X-tremeGENE HP DNA Transfection Reagent (Roche) in 10 cm dishes. The cells were collected 48 h post-transfection, washed twice with PBS, and subsequently resuspended in assay buffer (20 mM HEPES pH 7.5, 150 mM NaCl, and 0.1% Triton X-100 supplemented with Roche cOmplete Protease Inhibitor Cocktail). The cells were lysed by sonication and cleared by centrifugation for 5 min at 14,000 rpm and 4°C. The clarified lysates were flash frozen in liquid nitrogen and stored at –80°C until further use. DNA encoding CGI-99 (amino acids 1–244, 2–101, or 101–244) was inserted into pGEX-6p1 vector (Cytiva) to generate a fusion protein with an N-terminal glutathione S-transferase (GST) tag followed by a PreScission protease cleavage site. The fusion protein was expressed at 18°C overnight in BL21 (DE3) Rosetta2 *E. coli* cells upon induction with 0.2 mM IPTG. After harvesting, cells were lysed by sonication in binding buffer containing 20 mM Tris pH 7.5, 250 mM NaCl, and 1 mM DTT, supplemented with 0.1 µg ml$^{-1}$ pepstatin A and 100 µg ml$^{-1}$ AEBSF. The lysate was clarified by centrifugation for 30 min at 30,000 × *g* at 4°C and subsequently loaded onto Glutathione Sepharose 4B resin (Cytiva). The resin was washed with binding buffer and the protein was eluted with binding

buffer supplemented with 10 mM reduced glutathione (GSH, Sigma). Elution fractions containing the protein of interest were concentrated using a centrifugal filter (Amicon Ultra, MWCO 3 kDa, Sigma) before loading onto a size-exclusion Sepharose 200 column (Cytiva) and eluting with buffer containing 20 mM HEPES pH 7.5, 250 mM KCl, and 1 mM DTT. The protein-containing fractions were concentrated using a centrifugal filter to 1.4, 14.9, and 0.4 mg ml$^{-1}$ for CGI-99 constructs containing residues 1–244, 2–101, and 101–244, respectively. The samples were flash frozen in liquid nitrogen and stored at –80°C until further use. For pull-down experiments, 30 µg of each GST-CGI-99 construct was mixed with 500 µl of lysate from HEK293T cells transiently expressing one of the FAM98B constructs and incubated with 20 µl Glutathione Sepharose 4B resin (pre-equilibrated in assay buffer) at 4°C for 1 h. The resin was then washed with 5 × 1 ml assay buffer. The protein was eluted by the addition of 1 µl of 0.5 M reduced glutathione and 50 µl of 1× SDS loading dye. The samples were not heated prior to analysis on a 12% SDS-PAGE. The gels were scanned using the Typhoon FLA 9500 fluorescence scanner with the 473 nm laser and subsequently stained by Coomassie Blue.

## Chemical cross-linking and mass spectrometric (XL-MS) analysis

Purified tRNA-LC was diluted in cross-linking buffer (200 mM HEPES, pH 8.0, 150 mM KCl, 0.5 mM TCEP) to a final concentration of 1 mg ml$^{-1}$. To initiate the cross-linking reaction, 2 µl of a 25 mM stock of DSS (DSS-d$_0$/d$_{12}$, Creative Molecules) was added in independent replicates to two 50 µl aliquots of the complex, resulting in a final concentration of 1 mM DSS. Samples were incubated for 30 min at 25°C with mild shaking in an Eppendorf Thermomixer, and the reaction was quenched by addition of ammonium bicarbonate to a final concentration of 50 mM. Quenched samples were evaporated to dryness in a vacuum centrifuge and processed as described previously (*Leitner et al., 2014*). Briefly, disulfide bonds were reduced with TCEP, free thiols were alkylated by iodoacetamide, and proteins were sequentially digested with endoproteinase Lys-C (Wako, 1:100 enzyme-to-substrate ratio, 3 h at 37°C) and trypsin (Promega, 1:50, overnight at 37°C). Digested samples were desalted, purified by solid-phase extraction (SepPak tC18, Waters), and fractionated by SEC (Superdex Peptide PC 3.2/300, Cytiva) (*Leitner et al., 2014*). Two SEC fractions enriched in cross-linked peptides were collected and evaporated to dryness. The fractions were analyzed in duplicate by liquid chromatography-tandem mass spectrometry (LC-MS/MS) using a Thermo Easy nLC-1000 HPLC system coupled to a Thermo Orbitrap Elite mass spectrometer. The HPLC separation was performed on a Thermo Acclaim PepMap RSLC column (150 mm × 75 µm, 2 µm particle size) using the mobile phases A = water/acetonitrile/formic acid (98:2:0.15, v/v/v) and B = acetonitrile/water/formic acid (98:2:0.15, v/v/v), a gradient from 9% to 35% B in 60 min, and a flow rate of 300 nl/min. The mass spectrometer was operated in data-dependent acquisition mode with MS1 detection in the Orbitrap analyzer (120,000 resolution) and MS2 detection in the linear ion trap (normal resolution). For each cycle, the 10 most abundant precursor ions with a charge state of +3 or higher were selected for fragmentation in the linear ion trap with a normalized collision energy of 35%. Previously selected precursors were put on a dynamic exclusion list for 30 s.

## XL-MS data analysis

MS data were analyzed using xQuest (*Walzthoeni et al., 2012*), version 2.1.4. MS/MS spectra were searched against a custom database containing the target protein sequences and two contaminant proteins and a decoy database containing the reversed sequences. The main search parameters were enzyme = trypsin, maximum number of missed cleavages = 2, cross-linking sites = Lys and N terminus, fixed modification = carbamidomethylation of Cys, variable modification = oxidation of Met, MS1 mass tolerance = ± 15 ppm, MS2 mass tolerance = ± 0.2 Da for 'common' fragments and ±0.3 Da for 'cross-link' fragments. The scoring scheme of *Walzthoeni et al., 2012* was used. Post-search, the results were filtered to a mass error window of –6 to +3 ppm, a TIC value of 0.1% and a deltaS value of <0.9. MS/MS spectra of all remaining candidate hits were manually evaluated, and the false discovery rate at the non-redundant peptide pair level was adjusted to <5%. The list of cross-link peptide pair identifications from the two replicate experiments is provided in *Figure 2—source data 1* and *Figure 2—source data 2*.

## Limited proteolysis coupled with mass spectrometry

The tRNA-LC construct expressing N-terminally truncated DDX1 (RTCB:DDX1(436-740):FAM98B:CGI-99) was cloned, expressed, and purified as described for the full-length four-subunit tRNA-LC (RTCB:DDX1:FAM98B:CGI-99) construct except that the dialysis buffer was supplemented with 0.5 mM TCEP and the SEC buffer was supplemented with 1 mM DTT. The final concentration of the complex was 8.2 mg ml$^{-1}$. 1 mg of the complex at 1 mg ml$^{-1}$ was incubated with 2 μg of trypsin for 2.5 h at 20°C, after which AEBSF was added to a final concentration of 100 μg ml$^{-1}$. The resulting mixture was loaded onto two Superdex 200 10/300 columns (Cytiva) connected in series and eluted in 20 mM HEPES pH 8.0, 250 mM KCl, 1 mM DTT. As a reference, 0.4 mg of an untreated complex was run on the size-exclusion columns under the same conditions. Samples from the SEC peaks were analyzed by SDS-PAGE and stained by Coomassie Blue. The elution fractions corresponding to peaks A and B of the trypsin-treated RTCB:DDX1(436-740):FAM98B:CGI-99 were concentrated using a centrifugal filter (Amicon Ultra, Sigma, MWCO 30 kDa), and the components were identified by electrospray MS with a method accuracy of 0.5–1 Da.

## Truncation analysis of core tRNA-LC

The cloning and Sf9 cell expression of the truncated constructs was performed as described above for the deletion analysis of the tRNA-LC subunits with the following changes. The resulting plasmids encoded RTCB with an N-terminal His$_6$ tag followed by a TEV protease cleavage site; DDX1 with an N-terminal MBP tag followed by PreScission protease cleavage site, 3xFLAG tag and a second PreScission protease cleavage site; FAM98B with an N-terminal GFP tag followed by a TEV protease cleavage site; CGI-99 with an N-terminal mCherry tag followed by a TEV protease cleavage site. One half of clarified cell lysate (2 ml) was applied to 200 μl of Ni-NTA Superflow resin (QIAGEN) equilibrated with Ni wash buffer (20 mM HEPES pH 8.0, 150 mM NaCl, 15 mM imidazole, 0.1% Tween 20) in a gravity column. The resin was washed four times with 1 ml Ni wash buffer and bound proteins were eluted with 250 μl (E1) and 50 μl (E2) of Ni wash buffer-supplemented 250 mM imidazole. The second half of the clarified lysate was incubated for 2 h at 4°C with 20 μl of Anti-FLAG M2 magnetic beads (Sigma) equilibrated in FLAG wash buffer containing 20 mM HEPES pH 8.0, 150 mM NaCl, and 0.1% Tween 20. The beads were then washed with 3 × 500 μl FLAG wash buffer and eluted by the addition of 20 μl of 2× SDS loading dye. Samples of the clarified lysate (input), Ni-NTA, and FLAG elutions were analyzed by SDS-PAGE on AnykD Mini-PROTEAN TGX Precast Protein Gels (Bio-Rad). The samples were not heated before loading to preserve the fluorescence of the GFP and mCherry tags. The gels were visualized using a Typhoon FLA 9500 fluorescence scanner (Cytiva) with the 473 nm and the 532 nm laser to detect GFP and mCherry, respectively, and then stained with Coomassie Blue.

## Expression and purification of the minimal tRNA-LC

The cloning, Sf9 cell expression, and purification of the minimal tRNA-LC were performed as described above for the five-subunit tRNA-LC with the following changes. DNA encoding amino acids Ala696-Phe740 of human DDX1 was cloned into UC Berkeley MacroLab 438C (Addgene plasmid #55220). The resulting plasmid encoded untagged RTCB(1-505); DDX1(696-740) with an N-terminal His$_6$ tag followed by an MBP tag and a TEV protease cleavage site; untagged CGI-99(102-244); and FAM98B(200-239) with an N-terminal StrepII-tag followed by a GFP tag and a TEV protease cleavage site. For the first affinity step, the supernatant was incubated with 9 ml of Ni-NTA Superflow resin (QIAGEN), gently agitated at 4°C for 1 h. The solution was transferred to a glass column, and the matrix was washed and protein eluted as described above. The Strep wash buffer contained 20 mM HEPES pH 8.0, 500 mM NaCl, 0.1% Tween 20, 1 mM DTT, and the dialysis buffer contained 20 mM HEPES pH 8.0 and 150 mM KCl. The dialyzed protein mixture was concentrated using a centrifugal filter (Amicon Ultra, MWCO 10 kDa, Sigma), and the complex was further purified by SEC (Superdex 200 16/600, Cytiva), eluting with 20 mM HEPES pH 8.0, 150 mM KCl, 1 mM DTT. Peak fractions containing the minimal tRNA-LC subunits were pooled, concentrated to 3.6 mg ml$^{-1}$ using a centrifugal filter (Amicon Ultra, MWCO 10 kDa, Sigma), flash frozen in liquid nitrogen, and stored at –80°C.

## Expression and purification of CGI-99(2-101)

DNA encoding amino acids Phe2-Asp101 of human CGI-99 was cloned into the UC Berkeley MacroLab 1B vector (Addgene plasmid # 29653). The fusion protein containing an N-terminal His6 tag and a

TEV protease cleavage site was expressed in BL21 (DE3) Rosetta2 *E. coli* cells induced with 0.2 mM IPTG at 18°C overnight. The cells were harvested by centrifugation and lysed by sonication in lysis buffer: 20 mM Tris pH 8.0, 500 mM NaCl, 5 mM imidazole, supplemented with protease inhibitors AEBSF at 10 mg ml$^{-1}$, and Pepstatin A at 0.1 mg ml$^{-1}$. The lysate was cleared by centrifugation for 45 min at 20,000 × $g$ at 4°C. The supernatant was incubated with 15 ml HIS-Select Nickel Affinity Gel (Sigma) for 1 h at 4°C. The resin was transferred to a glass column and washed with 2 × 50 ml buffer containing 20 mM Tris pH 8.0, 500 mM NaCl, and 5 mM imidazole followed by 3 × 50 ml of the same buffer with the imidazole concentration raised to 10 mM. The protein was eluted with 5 × 5 ml of buffer containing 20 mM Tris pH 8.0, 500 mM NaCl, and 150 mM imidazole. The first four elution fractions were pooled, supplemented with 3 mg of His$_6$-tagged TEV protease, and dialyzed overnight at 4°C against buffer containing 20 mM Tris pH 8.0 and 250 mM KCl. The sample was applied to a 5 ml Ni-NTA Superflow cartridge (QIAGEN) in order to remove the uncleaved fusion protein and His$_6$-TEV protease. The flow-through fraction was collected, concentrated using a centrifugal filter (Amicon Ultra, MWCO 3 kDa, Sigma), and purified by SEC on a Superdex 75 26/600 column (Cytiva), eluting with 20 mM Tris pH 8.0, 250 mM KCl, and 1 mM DTT. CGI-99(2-101)-containing peak fractions were concentrated using a centrifugal filter (Amicon Ultra, MWCO 3 kDa, Sigma) to 24 mg ml$^{-1}$, flash frozen in liquid nitrogen, and stored at –80°C.

## Crystallization and structure determination of CGI-99(2-101)

Crystals of CGI-99(2-101) were obtained using the hanging drop vapor diffusion method at 4°C. 1 µl of protein solution containing CGI-99(2-101) at a concentration of 24 mg ml$^{-1}$ was mixed with 1 µl of reservoir solution containing 100 mM HEPES pH 7.5, 200 mM MgCl$_2$, and 18 or 20% isopropanol. After 21 days, the crystals were transferred into the reservoir solution supplemented with 30% hexylene glycol for cryoprotection. After 10 min, the crystals were flash frozen in liquid nitrogen. X-ray diffraction data were collected at beamline X06DA (PXIII) of the Swiss Light Source at the Paul Scherrer Institute in Villigen, Switzerland. The data were processed using XDS (*Kabsch, 2010*), and the space group was determined to be P6$_1$ using POINTLESS (*Evans, 2011*) with two copies in the asymmetric unit. The crystals diffracted to a resolution of 2.0 Å (native crystal) and 2.2 Å (S-SAD crystal). The structure was determined by a single-wavelength anomalous diffraction experiment utilizing endogenous sulfur atoms (S-SAD). Four 360° datasets were recorded at a wavelength of 2.0173 Å and merged using XSCALE (*Kabsch, 2010*). Seven 'S' sites were identified using Phenix.HySS (*Grosse-Kunstleve and Adams, 2003*), four of which corresponded to residues Cys19 and Cys69 (in both chains) and two corresponded to chloride ions coordinated by the protein. Phasing, density modification, preliminary model-building, and refinement were performed using Phenix.Autosol (*Terwilliger et al., 2009*). The model was further improved using Phenix.AutoBuild (*Terwilliger et al., 2008*), and the resulting structure was used as the search model for molecular replacement with the higher-resolution native dataset using Phenix.Phaser (*Adams et al., 2010*; *McCoy et al., 2007*). The model building was finished in Coot (*Emsley et al., 2010*) and refined using Phenix.Refine (*Afonine et al., 2012*). The final CGI-99(2-101) model contains residues 2–94 in chain A and residues 2–101 in chain B with three molecules of isopropanol, one molecule of hexylene glycol, and one chloride bound to each chain. An inter-molecular disulfide bond was observed between the Cys19 residues of the two chains in the asymmetric unit. Chain B was used as a template for structural superpositions using the DALI server (*Holm, 2020*).

## Expression and purification of RTCB

DNA encoding human RTCB was cloned into the UC Berkeley MacroLab 4B vector (gift from Scott Gradia, Addgene plasmid #30115). The fusion protein containing an N-terminal His$_6$ tag and a TEV protease cleavage site was expressed in Sf9 insect cells using the Bac-to-Bac baculovirus expression system (Invitrogen). Cells were infected at a density of 1 × 10$^6$ ml$^{-1}$, harvested 60 h after infection, and lysed by sonication in 20 mM HEPES pH 8.0, 150 mM NaCl, 0.1% Tween 20, supplemented with Roche cOmplete Protease Inhibitor Cocktail. The lysate was clarified by centrifugation for 30 min at 30,000 × $g$ at 4°C. The supernatant was applied to three 5 ml Ni-NTA Superflow cartridges (QIAGEN). The resin was washed with 20 mM HEPES pH 8.0, 500 mM NaCl, 0.1% Tween 20, and 10–20 mM imidazole, and bound protein was eluted with buffer containing 20 mM HEPES pH 8.0, 500 mM NaCl, 0.1% Tween 20, and 250 mM imidazole. The fusion tag was removed by a His$_6$-tagged TEV protease during

an overnight dialysis at 4°C against buffer containing 20 mM HEPES pH 8.0 and 500 mM NaCl. The dialyzed protein mixture was loaded onto the Ni-NTA Superflow cartridges, and the same procedure was used to remove the His$_6$-TEV protease, any remaining impurities, and uncleaved RTCB. RTCB-containing flow-through and wash fractions were pooled and concentrated using a centrifugal filter (Amicon Ultra, MWCO 10 kDa, Sigma), diluted with 20 mM HEPES pH 8.0 to a final NaCl concentration of 350 mM and reconcentrated. The protein was further purified by SEC (Superdex 200 16/600, Cytiva), eluting with 20 mM HEPES pH 8.0, 350 mM NaCl, 0.5 mM DTT. RTCB-containing fractions were pooled, concentrated to 17.5 mg ml$^{-1}$ using a centrifugal filter (Amicon Ultra, MWCO 10 kDa, Sigma), flash frozen in liquid nitrogen, and stored at –80°C.

## Crystallization and structure determination of RTCB

Crystals of GMP-bound RTCB were obtained using the hanging drop vapor diffusion method at 20°C. 0.5 μl of protein solution containing RTCB (7.7 mg ml$^{-1}$), GMP (0.5 mM), and an RNA oligonucleotide (0.15 mM, 5′-ACGUGCAAAGGCACUC-3′p) was mixed with 0.5 μl of reservoir solution (0.1 M sodium acetate pH 5.5, 0.6 M sodium formate, 14% [w/v] PEG 4K, and 5 mM CoCl$_2$). The crystals were transferred to the reservoir solution supplemented with 25% (v/v) glycerol for cryoprotection and then flash-cooled in liquid nitrogen. X-ray diffraction data were collected at beamline X06DA (PXIII) of the Swiss Light Source at the Paul Scherrer Institute in Villigen, Switzerland. The data were processed using XDS (*Kabsch, 2010*), and the space group was determined to be $P4_12_12$ using POINTLESS (*Evans, 2011*). The crystal diffracted to a resolution of 2.3 Å with two copies present in the asymmetric unit. The structure was solved by molecular replacement using Phenix.Phaser (*Adams et al., 2010*; *McCoy et al., 2007*) using the PhRtcB structure (PDB ID: 1uc2) as a search model. The initial model building was done using Phenix.Autobuild (*Terwilliger et al., 2008*) and finished manually in Coot (*Emsley et al., 2010*). The model was refined using Phenix.Refine (*Afonine et al., 2012*) with metal ion coordination not restrained and occupancy of metal ions fixed to 100%. The final RTCB-GMP model contains residues 3–55, 60–434, and 444–505; one molecule of GMP and two Co$^{2+}$ ions in chain A and residues 3–55, 59–440, 444–463, and 465–505; one molecule of GMP and two Co$^{2+}$ ions in chain B. Structural superpositions were performed using DALI pairwise alignment (*Holm, 2019*) with RTCB-GMP chain B as template.

## Data deposition

Atomic coordinates and structure factors for the reported crystal structures have been deposited with the Protein Data Bank under accession numbers 7P3A (CGI-99 N-terminal domain) and 7P3B (RTCB). XL-MS proteomics data have been deposited to the ProteomeXchange Consortium (http://proteomecentral.proteomexchange.org/cgi/GetDataset) via the PRIDE partner repository (*Perez-Riverol et al., 2019*) with the dataset identifier PXD025662.

## Acknowledgements

We are grateful to Beat Blattmann at the Protein Crystallization Center (University of Zurich) for assistance with initial crystallization experiments; Meitian Wang, Vincent Olieric, and Takashi Tomizaki at the Swiss Light Source (Paul Scherrer Institute, Villigen, Switzerland) for assistance with X-ray diffraction measurements; and the Functional Genomics Center Zurich for mass spectrometry analysis of the limited proteolysis samples. We thank Katja Bargsten for help with molecular cloning; Marcello Clerici for assistance with crystallographic data processing; members of the Martinez and Jinek groups for discussions and critical reading of the manuscript. This work was supported by Boehringer Ingelheim Fonds PhD Fellowship and the Forschungskredit program of the University of Zurich (grant no. FK-18-033) to AK, and the National Competence Center for Research (NCCR) RNA & Disease funded by the Swiss National Foundation. MJ and AL are members of the NCCR RNA & Disease. Research in the JM laboratory was funded by the Medical University of Vienna, Fonds zur Forderung der wissenschaftlichen Forschung (FWF) as Stand-Alone Projects (P29888) and through the RNA Biology Doctoral Program. MJ is an International Research Scholar of the Howard Hughes Medical Institute, Vallee Scholar of the Bert L&N Kuggie Vallee Foundation.

## Additional information

### Funding

| Funder | Grant reference number | Author |
|---|---|---|
| Boehringer Ingelheim Fonds | PhD Fellowship | Alena Kroupova |
| Schweizerischer Nationalfonds zur Förderung der Wissenschaftlichen Forschung | P29888 | Igor Asanović Stefan Weitzer Javier Martinez |
| University of Zurich | FK-18-033 | Alena Kroupova |
| Swiss National Science Foundation | | Martin Jinek Alexander Leitner |
| Medical University of Vienna | | Javier Martinez |
| Vallee Foundation | | Martin Jinek |

The funders had no role in study design, data collection and interpretation, or the decision to submit the work for publication.

### Author contributions

Alena Kroupova, Conceptualization, Formal analysis, Funding acquisition, Investigation, Methodology, Writing - original draft, Writing - review and editing; Fabian Ackle, Data curation, Investigation; Igor Asanović, Stefan Weitzer, Investigation, Writing - review and editing; Franziska M Boneberg, Alessia Chui, Investigation; Marco Faini, Formal analysis, Investigation; Alexander Leitner, Formal analysis, Writing - review and editing; Ruedi Aebersold, Formal analysis, Supervision, Writing - original draft; Javier Martinez, Conceptualization, Funding acquisition, Investigation, Methodology, Project administration, Supervision, Writing - review and editing; Martin Jinek, Conceptualization, Formal analysis, Funding acquisition, Investigation, Methodology, Project administration, Supervision, Writing - original draft, Writing - review and editing

### Author ORCIDs

Alena Kroupova ![ORCID] http://orcid.org/0000-0003-4166-1270
Fabian Ackle ![ORCID] http://orcid.org/0000-0002-7199-5004
Martin Jinek ![ORCID] http://orcid.org/0000-0002-7601-210X

### Decision letter and Author response

Decision letter https://doi.org/10.7554/eLife.71656.sa1
Author response https://doi.org/10.7554/eLife.71656.sa2

## Additional files

### Supplementary files

• Transparent reporting form

### Data availability

X-ray diffraction data and atomic models have been deposited in the Protein Data Bank under accession codes 7P3A (CGI-99 N-terminal domain) and 7P3B (RTCB in complex with GMP and Co(II)). The mass spectrometry proteomics data have been deposited to the ProteomeXchange Consortium (http://proteomecentral.proteomexchange.org) via the PRIDE partner repository with the dataset identifier PXD025662.

The following dataset was generated:

| Author(s) | Year | Dataset title | Dataset URL | Database and Identifier |
|---|---|---|---|---|
| Kroupova A, Jinek M | 2021 | N-terminal domain of CGI-99 | http://www.rcsb.org/structure/7P3A | RCSB Protein Data Bank, 7P3A |
| Kroupova A, Ackle F, Jinek M | 2021 | Human RNA ligase RTCB in complex with GMP and Co(II) | http://www.rcsb.org/structure/7P3B | RCSB Protein Data Bank, 7P3B |
| Leitner A, Faini M, Aebersold R | 2021 | tRNA-LC cross-linking/mass spectrometry data | http://proteomecentral.proteomexchange.org/cgi/GetDataset?ID=PXD025662 | ProteomeXchange, PXD025662 |

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
