## [Editor Report]

Kroupova and colleagues present the manuscript ‘Molecular architecture of human tRNA ligase complex,’ which describes the first detailed dissection of the structure and assembly of the essential, multi-subunit tRNA ligase complex. The authors present, alongside crystal structures of the Rtcb catalytic subunit and the N-terminus of the associated CGI-99 subunit, a comprehensive deletion analysis and mass spectrometry investigation of the composition and assembly of the entire hetero-oligomeric assembly, identifying a novel subassembly between the CGI-99 and FAM98B subunits. The study is elegant, beautifully presented and written, easily followed, and interesting, providing a high-quality and important dissection of this essential complex.

---

## [Decision Letter]

**Decision letter after peer review:**

Thank you for submitting your article "Molecular architecture of the human tRNA ligase complex" for consideration by *eLife*. Your article has been reviewed by 3 peer reviewers, and the evaluation has been overseen by a Timothy Nilsen as Reviewing Editor and Cynthia Wolberger as the Senior Editor. The following individuals involved in review of your submission have agreed to reveal their identity: Alessandro Vannini (Reviewer #3); Oliver Weichenrieder (Reviewer #4).

Essential revisions:

All of the reviewers and the reviewing editor found this to be a significant contribution that in principle is suitable for *eLife*. All of the reviewers have made suggestions regarding presentation of the work. Please address these as thoroughly as possible. In addition, it was the general consensus that the metal specificity experiments suggested by reviewer 2 would add substantively to the paper. Although not required, we suggest that these experiments be performed. If you decide that you would prefer not to do the experiments, the metal specificity should at least be discussed.

*Reviewer #2:*

The authors present a technically accomplished multi-modal study of the human tRNA ligase complex entailing inter- and intra-subunit contact mapping by crosslinking/mass spec, protein truncations to delineate sufficiency for protein-protein interactions, a crystal structure of the N-domain of the CGI-99 subunit, delineation of a minimal active ligase complex, and a crystal structure of the RTCB ligase subunit.

The work is clearly of general interest. It provides a foundation for future efforts to solve the structure of the entire tRNA ligase complex.

Two significant scientific issues need to be addressed as part of this study that will require further experimentation (that can be accomplished readily).

1) The authors underplay the significance of the ligation assays presented in Figure 1 supplemental panel B, which show that the recombinant 5-subunit (holo) and 4-subunit (core) ligase complexes have feeble strand joining activity compared to the minimal complex containing RTCB and fragments of other core subunits. Indeed, the minimal complex is just as active as RTCB alone (all in the presence of archease). This is a remarkable finding that needs to be discussed and placed in a biological context. Is there a reason for limiting the activity of RTCB by encumbering it in the context of the tRNA ligase complex? Or perhaps the results are a function of the use of an "artificial" substrate in the ligation assays, which score for intramolecular circularization of a 21-mer RNA. I would consider a scenario in which the other components of the complex impart specificity to RtcB for broken tRNA substrates over non-specific RNA with ligatable ends. This is exactly the case for T4 Rnl1, which is 100-fold better at sealing a broken tRNA than at intramolecular circularization (Wang LK et al., 2007 RNA). The upshot is that the authors ought to assay in this study the sealing of a pre-cleaved tRNA substrate by the various forms of the ligase interrogated in supplemental panel B. (They have deployed such assays in their previous studies, so it shouldn't be a burden to do so here.)

2) The structure of human RTCB is of a complex with two cobalt ions in the active site. I think it is generally accepted that RtcB enzymes require manganese as the divalent cation cofactor. (The ligation assays in the present study are performed in the presence of manganese). The authors note differences in metal coordination geometry between their cobalt structure and previous manganese structures of PhoRtcB, and they suggest that the geometries might be dynamic during the several chemical steps of the RtcB ligation pathway. The critical issue here is whether a cobalt complex is actually off-pathway, insofar as extensive studies of *E. coli* RtcB have shown that cobalt is unable to serve as a cofactor for (i) RtcB reaction with GTP to form the RtcB-GMP intermediate; (ii) GMP transfer to a polynucleotide 3'-phosphate to form the polynucleotide 3'ppG intermediate; or (iv) attack of a 5'-OH strand on the 3'ppG end to complete the ligation reaction. Moreover, and most important, cobalt is inhibitory to each of these reactions in the presence of manganese. These data are most simply explained by cobalt binding to one or both metal sites in a non-productive off-pathway state. Thus, it is critical for the authors to present in this study an experiment that tests whether cobalt can support human RTCB ligation and whether it inhibits in the presence of manganese.

*Reviewer #3:*

Kroupova and colleagues present the manuscript "Molecular Architecture of human tRNA ligase complex" which describes the first detailed dissection of the structure and assembly of the essential, multi-subunit tRNA ligase complex. The authors present, alongside crystal structures of the Rtcb catalytic subunit and the N-terminus of the associated CGI-99 subunit, a comprehensive deletion analysis and mass spectrometry investigation of the composition and assembly of the entire hetero-oligomeric assembly, identifying a novel sub-assembly between the CGI-99 and FAM98B subunits. The study is elegant, beautifully presented and written, easily followed and interesting, providing a high-quality and important dissection of this essential complex.

*Reviewer #4:*

The tRNA ligase complex participates in protein (endonuclease and ligase)-mediated RNA splicing, which contrasts with the better known RNA-mediated splicing in the context of the spliceosome. Protein-mediated RNA splicing may date back to the ancient RNA-protein world where it might have served to defend RNA against invading introns. RTCB-like RNA ligases such as found in the human tRNA ligase complex act in numerous cellular pathways including bacterial RNA repair, and they follow a highly interesting and complex but structurally poorly understood mechanism that joins RNA 5'-hydroxyl ends with 3'-ends that carry a 2'-3' cyclic phosphate. Although the components of the human tRNA ligase complex are known and although there is crystal structural information on archaeal RTCB homologs, the function of the non-enzymatic components and the structural organization of the complex are unknown.

The quality of the data in Kroupova et al., and their presentation in the manuscript is outstanding, which is easy to follow even for general readers. The authors carefully avoid to over-interpret their cross-linking data, although some remarks on this as outlined in my detailed remarks may benefit readers who are less familiar with this method. The methods are described with exceptional detail and information and will be extremely useful to scientists who plan similar approaches for their own protein complexes of choice and especially if these are still too undefined and/or flexible to be amenable for direct structural analysis by X-ray crystallography or single-particle cryo-electron microscopy.

This is a prime example for an analytic biochemical approach with modern methodology to a challenging problem in structural biology.

---

## [Author Response]

Essential revisions:All of the reviewers and the reviewing editor found this to be a significant contribution that in principle is suitable for eLife. All of the reviewers have made suggestions regarding presentation of the work. Please address these as thoroughly as possible. In addition, it was the general consensus that the metal specificity experiments suggested by reviewer 2 would add substantively to the paper. Although not required, we suggest that these experiments be performed. If you decide that you would prefer not to do the experiments, the metal specificity should at least be discussed.

We thank the Reviewing Editor and the Reviewers for their positive evaluations of our study and for their constructive criticisms. We have done our best to address these comments by generating additional biochemical data to support our structural observations and aid in their interpretation, and making additional textual revisions in the manuscript. We believe that these changes have strengthened the overall conclusions of our study and improved the clarity of the manuscript to warrant its publication.

Reviewer #2:The authors present a technically accomplished multi-modal study of the human tRNA ligase complex entailing inter- and intra-subunit contact mapping by crosslinking/mass spec, protein truncations to delineate sufficiency for protein-protein interactions, a crystal structure of the N-domain of the CGI-99 subunit, delineation of a minimal active ligase complex, and a crystal structure of the RTCB ligase subunit.The work is clearly of general interest. It provides a foundation for future efforts to solve the structure of the entire tRNA ligase complex.Two significant scientific issues need to be addressed as part of this study that will require further experimentation (that can be accomplished readily).1) The authors underplay the significance of the ligation assays presented in Figure 1 supplemental panel B, which show that the recombinant 5-subunit (holo) and 4-subunit (core) ligase complexes have feeble strand joining activity compared to the minimal complex containing RTCB and fragments of other core subunits. Indeed, the minimal complex is just as active as RTCB alone (all in the presence of archease). This is a remarkable finding that needs to be discussed and placed in a biological context. Is there a reason for limiting the activity of RTCB by encumbering it in the context of the tRNA ligase complex? Or perhaps the results are a function of the use of an "artificial" substrate in the ligation assays, which score for intramolecular circularization of a 21-mer RNA. I would consider a scenario in which the other components of the complex impart specificity to RtcB for broken tRNA substrates over non-specific RNA with ligatable ends. This is exactly the case for T4 Rnl1, which is 100-fold better at sealing a broken tRNA than at intramolecular circularization (Wang LK et al., 2007 RNA). The upshot is that the authors ought to assay in this study the sealing of a pre-cleaved tRNA substrate by the various forms of the ligase interrogated in supplemental panel B. (They have deployed such assays in their previous studies, so it shouldn’t be a burden to do so here.)

We thank the Reviewer for the comments. We agree that the observation of reduced activity of the fulllength 5- and 4-subunit tRNA-LC complexes when compared with RTCB alone or the minimal tRNALC (Figure 1—figure supplement 1) is intriguing. One the one hand, this could suggest that activity of the 5- or 4-subunit complexes is autoinhibited, and thus potentially regulated, by the non-catalytic subunits via protein regions or domains that are not involved in complex assembly. On the other, this finding could also be explained by a limited stability of the holo- and core complexes under the conditions employed for the ligase activity assay, resulting in reduced specific activities of these species. For this reason, we would prefer to remain cautious so as not to overinterpret the results. We mention this result explicitly on p. 9, lines 241-248, and briefly discuss its potential significance in the Discussion section.

Autoinhibition of the tRNA-LC would potentially add another layer to its regulation, alongside the recently reported sensitivity to oxidative stress. We tested whether the autoinhibition can be relieved in the presence of ATP to probe whether the ATPase activity of DDX1 is possibly involved. However, the addition of ATP had no effect under the conditions tested, suggesting that DDX1 does not play a role. Although the results shown in Figures S1B and S1C might represent a single snapshot of a more complex regulatory landscape, we believe that a more extensive analysis of the tRNA-LC activity and its regulation falls outside of the scope of the present manuscript.

Regarding the choice of the substrate, we agree with the Reviewer that it might conceivably impact the experimental outcome, although we have previously shown that the human tRNA-LC is quite nonselective with respect to RNA substrates. We thus chose a cautious interpretation of the presented data in this manuscript, which we would aim to use as a starting point for a future comprehensive study of the regulation of the tRNA-LC aided by our current and ongoing structural work.

2) The structure of human RTCB is of a complex with two cobalt ions in the active site. I think it is generally accepted that RtcB enzymes require manganese as the divalent cation cofactor. (The ligation assays in the present study are performed in the presence of manganese). The authors note differences in metal coordination geometry between their cobalt structure and previous manganese structures of PhoRtcB, and they suggest that the geometries might be dynamic during the several chemical steps of the RtcB ligation pathway. The critical issue here is whether a cobalt complex is actually off-pathway, insofar as extensive studies of *E. coli* RtcB have shown that cobalt is unable to serve as a cofactor for (i) RtcB reaction with GTP to form the RtcB-GMP intermediate; (ii) GMP transfer to a polynucleotide 3’-phosphate to form the polynucleotide 3’ppG intermediate; or (iv) attack of a 5’-OH strand on the 3’ppG end to complete the ligation reaction. Moreover, and most important, cobalt is inhibitory to each of these reactions in the presence of manganese. These data are most simply explained by cobalt binding to one or both metal sites in a non-productive off-pathway state. Thus, it is critical for the authors to present in this study an experiment that tests whether cobalt can support human RTCB ligation and whether it inhibits in the presence of manganese.

We agree with the Reviewer that the question of the divalent metal selectivity of human RTCB is important for the correct interpretation of the structural data. To address this issue, we have carried out additional ligase activity assays to determine the selectivity of RTCB and the full-length, 4-subunit tRNA-LC, and present the results in Figure 6 —figure supplement 1 and describe them on p. 10, lines 252-255. Unlike *E. coli* RtcB, human RTCB is highly active in the presence of either Mn^2+^ or Co^2+^ ions, and detectably active in the presence of Mg^2+^ and Zn^2+^. These results would suggest that the Co^2+^ complex captured in the crystal structure depicts an on-pathway state in the catalytic mechanism of RTCB. It is currently unclear whether Mn^2+^ or Co^2+^ is the physiologically relevant ion in vivo, and we cannot exclude the possibility that the two active site positions are occupied by different divalent cations under physiological conditions (e.g. Mg^2+^ in site A and Mn^2+^ or Co^2+^ in site B), as we note in the Discussion section of the manuscript (p. 13-14, lines 341-357). Despite extensive efforts, we have so far been unable to grow RTCB crystals from a crystallization solution lacking divalent metals, so as to be able to perform metal soaking experiments. We agree that further experiments and experimental approaches, for example inductively coupled plasma mass spectrometry (ICP-MS) analysis of natively-purified tRNA-LC, will be needed to characterize the divalent metal requirements of human RTCB and tRNA-LC but we believe that these studies fall outside of the scope of the current manuscript.